# Single [0001]-oriented zinc metal anode enables sustainable zinc batteries

Xiaotan Zhang[1,5], Jiangxu Li [2,3,5], Yanfen Liu[1], Bingan Lu [4], Shuquan Liang[1] ✉ & Jiang Zhou [1] ✉

The optimization of crystalline orientation of a Zn metal substrate to expose more Zn(0002) planes has been recognized as an effective strategy in pursuit of highly reversible Zn metal anodes. However, the lattice mismatch between substrate and overgrowth crystals has hampered the epitaxial sustainability of Zn metal. Herein, we discover that the presence of crystal grains deviating from [0001] orientation within a Zn(0002) metal anode leads to the failure of epitaxial mechanism. The electrodeposited [0001]-uniaxial oriented Zn metal anodes with a single (0002) texture fundamentally eliminate the lattice mismatch and achieve ultra-sustainable homoepitaxial growth. Using high-angle angular dark-filed scanning transmission electron microscopy, we elucidate the homoepitaxial growth of the deposited Zn following the "~ABABAB~" arrangement on the Zn(0002) metal from an atomic-level perspective. Such consistently epitaxial behavior of Zn metal retards dendrite formation and enables improved cycling, even in Zn||NH$_4$V$_4$O$_{10}$ pouch cells, with a high capacity of 220 mAh g$^{-1}$ for over 450 cycles. The insights gained from this work on the [0001]-oriented Zn metal anode and its persistently homoepitaxial mechanism pave the way for other metal electrodes with high reversibility.

Zinc (Zn) dendrite growth poses a major hurdle in the advancement of Zn-metal batteries, leading to reduced Coulombic efficiency (CE) and short circuits[1–5]. To realize planarized Zn deposition, tailoring the crystalline orientation of Zn metal anodes to expose the closest-packed Zn(0002) planes has attracted substantial research interests, owing to their high corrosion resistance and low surface energy[6–8]. Several methodologies have been demonstrated to achieve this hexagonal Zn texture, including constructing a pseudomorphic substrate to provide a tolerable lattice misfit (generally <18%) with the Zn(0002) plane[9,10], optimizing the electrolyte to direct the generation of an optimum dense surface[11], and reconstructing Zn metal matrix to regulate its crystallographic orientation[12–14]. Despite these considerable efforts, the attainment of single (0002)-textured Zn metal anodes that can maintain sustainably epitaxial growth still faces challenging. Furthermore, the growth mechanism and critical determinant responsible

for continuous epitaxy reported in literatures remain either deficient or ambiguous, necessitating further exploration and investigation.

Epitaxial growth is a process that involves the synthesis of crystalline materials with identical structure and orientation as the substrate surface[15,16]. This process is significantly influenced by the chemical potential between the substrate and overgrowth crystals[17]. Epitaxial growth can occur through two types: heteroepitaxy and homoepitaxy[18,19]. In heteroepitaxy, the distinct chemical potential arises from differences in chemical bonding and lattice orientation of the epitaxial interface[20], whereas homoepitaxy merely requires consideration of lattice mismatch[21,22], making it more favorable for achieving sustainably epitaxial growth. According to Royer's theory[23], the lattice mismatch ($f$) between the overgrowth and substrate crystals is a critical factor in determining epitaxial growth, with the epitaxial orientation of the former dependent on the crystallographic

[1]School of Materials Science and Engineering, Central South University, Changsha 410083 Hunan, PR China. [2]Department of Applied Chemistry, University of Science and Technology of China, Hefei 230026 Anhui, PR China. [3]Shenyang National Laboratory for Materials Science, Institute of Metal Research, Chinese Academy of Sciences, Shenyang 110016, PR China. [4]School of Physics and Electronics, Hunan University, Changsha 410082 Hunan, PR China. [5]These authors contributed equally: Xiaotan Zhang, Jiangxu Li. ✉e-mail: lsq@csu.edu.cn; zhou_jiang@csu.edu.cn

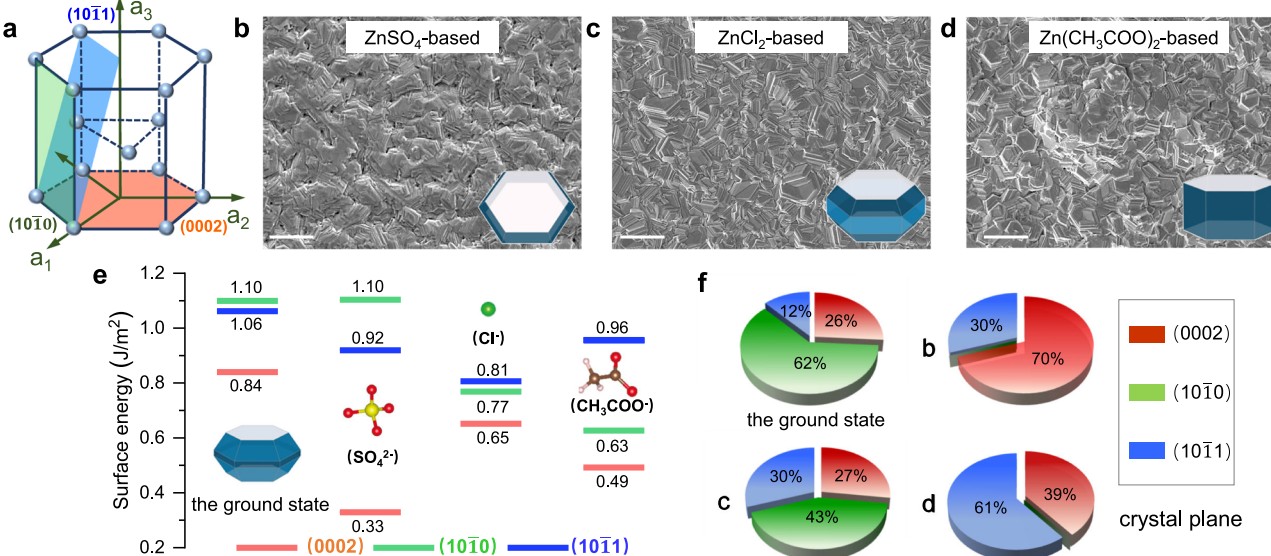

**Fig. 1 | Anion-induced texture regulation of Zn crystals and preparation of Zn metal electrodes. a** The illustration of the hexagonal close packed (hcp) structure of Zn. SEM images of the electrodeposited Zn metals in (**b**) ZnSO₄, (**c**) ZnCl₂, and (**d**) Zn(CH₃COO)₂ systems, respectively (scale bar, 20 μm). The inset images correspond to the equilibrium shape of Zn crystals in different system, obtained from HSE06 calculations. **e** Surface energies of Cl⁻, SO₄²⁻ and CH₃COOH⁻ on Zn(10$\bar{1}$0), Zn(10$\bar{1}$1) and Zn(0002) planes, respectively. Colors for different atoms: white for H, brown for C, red for O, yellow for S and green for Cl. **f** The ratios of different crystal planes (Zn(10$\bar{1}$0), Zn(10$\bar{1}$1) and Zn(0002)) in the ground state and the electrodeposited Zn crystals of **b** to **c**.

orientation of the latter. Despite the limited occurrence of grain with distinct crystalline orientations within the substrate crystal, the overgrowth crystal must undergo uniform strain to match the substrate crystal, resulting in the generation of misfit dislocations[24]. As lattice strain progressively accumulates, it eventually surpasses the critical thickness ($t_c$) of the epitaxial crystal, leading to the failure of epitaxial mechanism. The $t_c$ for the overgrowth crystal can be related to the lattice mismatch by $t_c = b(f_s/f)^2$, where $f_s$ represents the stability threshold in the one-dimensional model of the epitaxial interface, $f$ is only dependent on the lattice parameter ($a$ and $b$) of the substrate and the epitaxial layer ($f = (b-a)/a$)[25,26]. To achieve ultra-sustainable epitaxial growth of Zn(0002) metal anodes, two essential criteria must be met: homoepitaxy and single crystalline orientation. Homoepitaxy ensures chemical congruence between the substrate and the overgrowth crystals, allowing for the disregard of chemical disparities[27]. Single crystalline orientation results in a near-zero lattice mismatch at the epitaxial interface, which in turn facilitates the prolonged growth of homoepitaxial layers. Therefore, it is highly imperative to construct Zn metal anodes with a single (0002) texture to constrain the crystalline orientation for planar Zn electrodeposition[28].

In this work, we proposed and designed a [0001]-uniaxial oriented Zn metal anode, which possesses a single Zn(0002) texture. This Zn(0002) metal anode exhibits ultra-sustainable epitaxial growth and low inner stress, even under high depth of discharge (DOD) and/or high areal capacity. With the aid of high-angle angular dark-filed scanning transmission electron microscopy (HAADF-STEM), we observed that Zn atoms grow layer-by-layer following a "~ABABAB~" sequence of closely packed (0002) planes. Furthermore, we performed a homoepitaxial experiment on the Zn(0002) metal anode, characterized by a predominant (0002) texture and a minor presence of other Zn textures. Our findings suggest that owing to the existence of lattice mismatch at the epitaxial interface, once the deposited Zn exceeds the critical thickness of epitaxial layer, subsequently grown Zn becomes random and disordered and eventually forms Zn dendrites. These results emphasize the significance of maintaining a single (0002) texture to achieve sustainable homoepitaxy and suppress dendrite in Zn metal electrode. As expected, the Zn(0002) metal anode offers enhanced cycling when assembled with NH₄V₄O₁₀

cathode in pouch cells, exhibiting a remarkable 80% capacity retention over 450 cycles.

## Results

### Fabrication and characterization of [0001]-oriented Zn metal electrodes

Figure 1a illustrates a hexagonal closed-packed structure of Zn unit cell, where the basal surface corresponds to the Zn(0002) plane. The Zn(0002) plane has a lower surface energy of 0.84 J m⁻² in comparison to the Zn(10$\bar{1}$0) and Zn(10$\bar{1}$1) planes, confirming its high thermodynamic stability[29,30]. We utilized direct-current electrodeposition in boric acid (H₃BO₃)/Zn-based solutions to fabricate Zn metal electrodes with distinctive crystal morphology on a copper (Cu) foil. This was achieved through the incorporation of H₃BO₃ additive and vigorous stirring, creating an ideal environment to eliminate harmful side reactions[31–35]. As a result, under the prerequisite of mitigating side reactions, the crystallographic orientation and texture of the electrodeposited Zn metals were exclusively dependent on the anions present in the deposited electrolytes. Detailed experimental parameter regulation is provided in Supplementary Table 1 and Supplementary Figs. 1–3. The Zn metals deposited from ZnCl₂ and Zn(CH₃COO)₂ systems displayed irregular crystalline orientation, whereas a single (0002)-textured Zn metal was obtained from the ZnSO₄ system, as shown in the scanning electron microscopy (SEM) images (Fig. 1b–d).

To elucidate the role of anions in directing the crystal growth of deposited Zn metals, the surface energies (Fig. 1e) of low-index crystal planes (including Zn(10$\bar{1}$0), Zn(10$\bar{1}$1), and Zn(0002)) were calculated in these three different systems: ZnCl₂, ZnSO₄, and Zn(CH₃COO)₂. Selective adsorption of anions resulted in a reduction in the surface energies across all crystal planes. Notably, the surface energy of Zn(0002) plane in the ZnSO₄ system experienced a sharp drop, suggesting a preference for Zn(0002) plane exposure in the presence of SO₄²⁻ anions. Based on the Gibbs-Wulff theory of crystal growth, crystals with lower surface energy exhibit faster lateral growth rates, consequently emerging as the primary exposed crystal planes[36]. The equilibrium shapes of the prepared Zn metals, derived by a Wulff construction[37] in these three systems, had different crystal morphologies (inset of Fig. 1b–d). Theoretical calculations indicate that varying

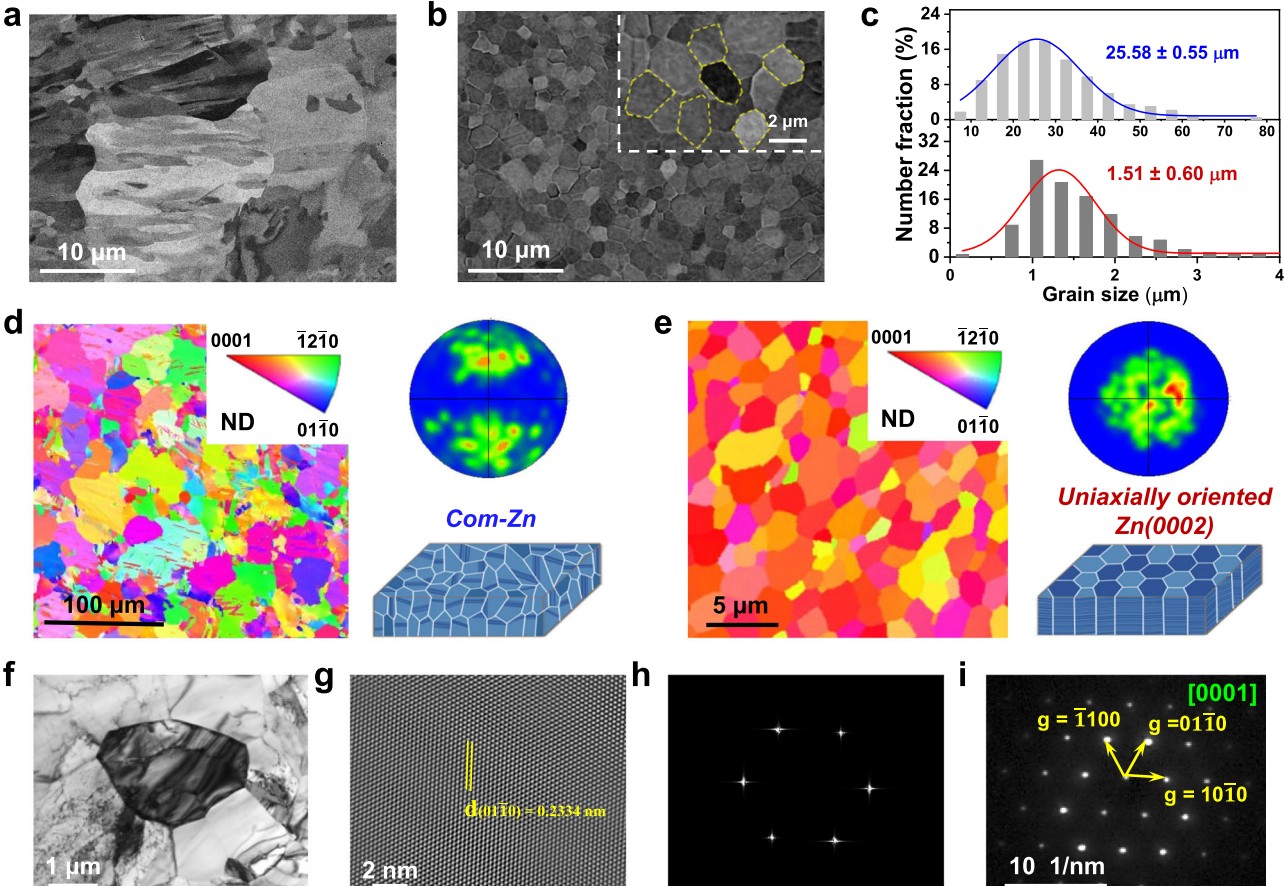

**Fig. 2 | Microstructural characterizations of uniaxially oriented Zn(0002) metal.** SEM images of (**a**) the com-Zn and (**b**) electrodeposited Zn(0002) metals with an magnified image of hexagonal Zn grains (inset). **c** Grain size distributions from statistical SEM measurements for the com-Zn and the electrodeposited Zn(0002) metals. 3D illustration, plane-view EBSD maps and corresponding (0002) pole figures of (**d**) the com-Zn and (**e**) electrodeposited Zn(0002) metals. Detailed crystallographic observation of the electrodeposited Zn(0002) metal: **f** Typical bright-field TEM image, **g** corresponding high-resolution TEM image, **h** FFT pattern and **i** 4D-STEM nanobeam diffraction pattern of **f**.

the anions of deposited solution gave rise to crystal texture changes (Fig. 1f). Specifically, it is evident that the ratio of Zn(0002) plane in the ZnSO$_4$ system was significantly higher than that in both ZnCl$_2$ and Zn(CH$_3$COO)$_2$ systems, aligning with X-ray diffraction (XRD) patterns (Supplementary Fig. 4).

Considering the universality and potential for large-scale application of the electrodeposited Zn(0002) metal, we here used commercial Zn (com-Zn)[38] metal as a control sample in the whole article. The microstructure of both com-Zn and Zn(0002) metals was systematically investigated through XRD, SEM, electron backscatter diffraction (EBSD), and transmission electron microscopy (TEM) observations. In contrast to the single (002) characterized peak of Zn(0002) metal, the XRD pattern (Supplementary Fig. 5) of the com-Zn metal displayed multiple peaks, implying a disordered crystalline orientation. Additionally, the residual stresses of Zn(0002) metal (14.8 MPa) were found to be lower than that of com-Zn metal (22.8 MPa), indicating minimal lattice strain and misfit dislocations[9,39]. Furthermore, the com-Zn metal demonstrated a random and irregular distribution of crystal grains with an average grain size value of 25.58 ± 0.55 μm (Fig. 2a and Supplementary Fig. 6a), which was further confirmed by the EBSD characterization (Fig. 2d and Supplementary Fig. 7a). Conversely, the Zn(0002) metal showed numerous equiaxed grains with an average grain size of 1.51 ± 0.60 μm (Fig. 2b and Supplementary Fig. 6b, c). The plane view EBSD maps and pole figures (Fig. 2e and Supplementary Fig. 7b) revealed that the Zn(0002) metal presented a single <0001> texture, showing a locked crystalline orientation. This result was further demonstrated by TEM (Fig. 2f), fast

fourier transform (FFT), and selected area electron diffraction (SAED) characterizations. The diffraction spots of the Zn atoms perfectly matched with the Zn(0002) lattice plane and the grain was in the [0001] orientation (Fig. 2h, i), where the corresponding interplanar spacing of 0.2334 nm was determined to the (01$\bar{1}$0) plane of Zn[40] (Fig. 2g). These comprehensive characterizations of the crystal structure properly indicate that the electrodeposited Zn(0002) metal has only one [0001]-crystallographic axis.

## Ex situ morphological and structural evolution of Zn plating/stripping

Zn metal shows a preference for forming disordered and nonplanar platelet electrodeposits at liquid-solid interfaces due to its lower thermodynamic free energy[41–45]. The Zn plating process can be divided into two stages: nucleation and continuous growth, which are influenced by the current density in accordance with the Bulter-Volmer electrode kinetics relationship[46]. Notably, the Zn(0002) metal displayed a lower nucleation overpotential ($\eta_n$) than that of com-Zn metal, irrespective of current density, whereas the plateau overpotential ($\eta_p$) exhibited the opposite trend (Supplementary Fig. 8a, b). This result suggests that Zn(0002) metal can suppress dendrite growth effectively in comparison with com-Zn metal[47].

Besides, to investigate the significance of a Zn metal electrode possessing a single (0002) texture for sustaining epitaxial growth, we constructed Zn metal electrodes with a predominant (0002) texture and a minor presence of other Zn textures as an intermediate state (IMS-Zn(0002)) between com-Zn and single (0002)-textured Zn

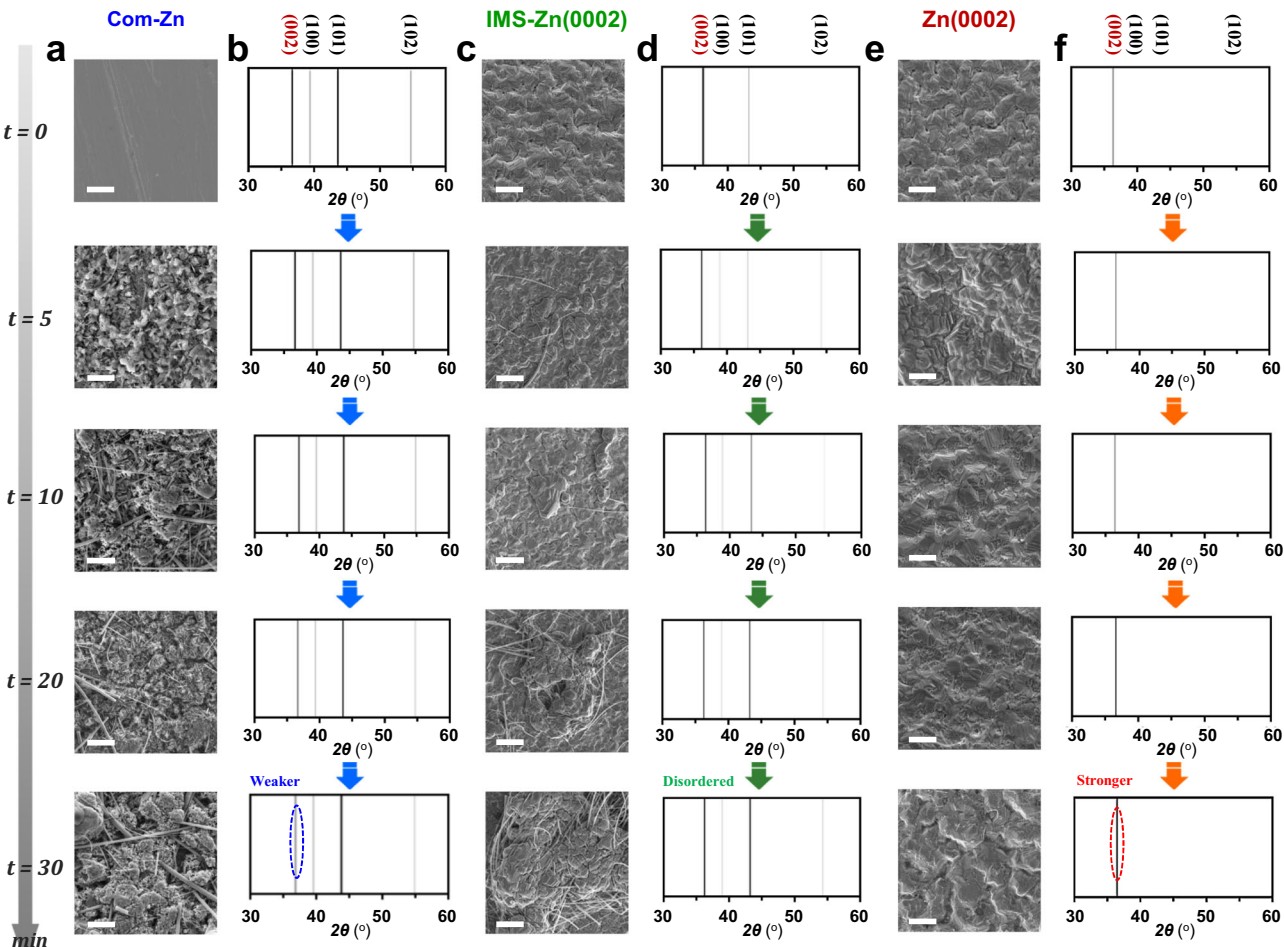

**Fig. 3 | Morphologies and orientation of Zn plating and stripping.** SEM images of (**a**) com-Zn, (**c**) IMS-Zn(0002) and (**e**) Zn(0002) electrodes in Zn||Zn cells after plating 0 to 30 min (scale bar, 20 μm). XRD patterns of (**b**) com-Zn, (**d**) IMS-Zn(0002) and (**f**) Zn(0002) electrodes after plating from 0 to 30 min. Current density, $J = 4$ mA cm$^{-2}$.

electrodes. The IMS-Zn(0002) metal electrode, featuring an average grain size value of $8.40 \pm 0.30$ μm, showed a hexagonal morphology similar to that of the Zn(0002) metal electrode (Supplementary Fig. 9a and Supplementary Fig. 10a, b). The XRD pattern (Supplementary Fig. 9b) of the IMS-Zn(0002) electrode displayed a highly pronounced (002) peak and a weak (101) peak. The relative texture coefficient (RTC) of each lattice plane was calculated using the following formula[48],

$$RTC_{(hkl)} = \frac{I_{(hkl)}/\bar{I}_{(hkl)}}{\sum[I_{(hkl)}/\bar{I}_{(hkl)}]} \times 100 \qquad (1)$$

Where, $I_{(hkl)}$ represents the intensity obtained from textured Zn sample, and $\bar{I}_{(hkl)}$ is the intensity of the standard oriented Zn sample (from JCPDS data). The corresponding RTC$_{(002)}$ of com-Zn, IMS-Zn(0002) and Zn(0002) electrodes was 41, 93 and 100, respectively (Supplementary Fig. 11). This result indicates that the IMS-Zn(0002) metal electrodes had only a minor (10$\bar{1}$1) texture predominantly featuring the (0002) texture, which was further confirmed by the EBSD characterization (Supplementary Fig. 12a–d).

The morphology evolution of com-Zn, IMS-Zn(0002) and Zn(0002) metal electrodes during plating and stripping was characterized by a series of in situ and ex situ tests. On the stripping side, both com-Zn and IMS-Zn(0002) electrodes exhibited numerous randomly-sized pores, which became more pronounced with discharge time (Supplementary Fig. 13a, b). Corrosion by-products were even formed on the com-Zn surface. In contrast, the Zn(0002) surface

displayed greater corrosion resistance and a more ordered stripping process (Supplementary Fig. 13c). On the plating side, scattered and loose Zn deposits were observed on the com-Zn surface, which rapidly grew along the separator direction (Fig. 3a). The IMS-Zn(0002) electrode maintained a relatively flat deposition surface during the 10 min plating. However, with an increase in deposition time, disordered and uneven Zn deposits formed on the IMS-Zn(0002) surface (Fig. 3c). In contrast, hexagonal-Zn deposition spread parallelly without Zn dendrites on the Zn(0002) surface during plating (Fig. 3e). This phenomenon was further affirmed through in situ optical microscopic analysis in transparent Zn||Zn symmetrical cells (Supplementary Fig. 14a, b). Even after 1 h plating, flat and densely packed Zn deposition was observed on the Zn(0002) surface using confocal laser microscope (LSM, Supplementary Fig. 15a–c), with a lower surface roughness (0.3 μm) than on com-Zn (2.8 μm) and IMS-Zn(0002) (1.9 μm) surfaces. Ex situ XRD was conducted to detect the structural evolution of Zn metal electrodes during the charging process, as shown in Fig. 3b, d, f and Supplementary Fig. 16a–c. The peaks located at approximately 36° in both electrodes were assigned to the (0002) plane of Zn crystal[49]. The XRD patterns showed that the deposited Zn on com-Zn and IMS-Zn(0002) surfaces lacked a fixed orientation, while a single (002) peak was consistently observed using the Zn(0002) electrode from the beginning to the end, illustrating ultra-sustainable epitaxial growth between the overgrowth and the substrate crystals. Furthermore, there was a significant increase in residual stress for both IMS-Zn(0002) (from 16.5 MPa to 60.8 MPa) and com-Zn (from 22.8 MPa to 110.7 MPa) metal electrodes before and after deposition

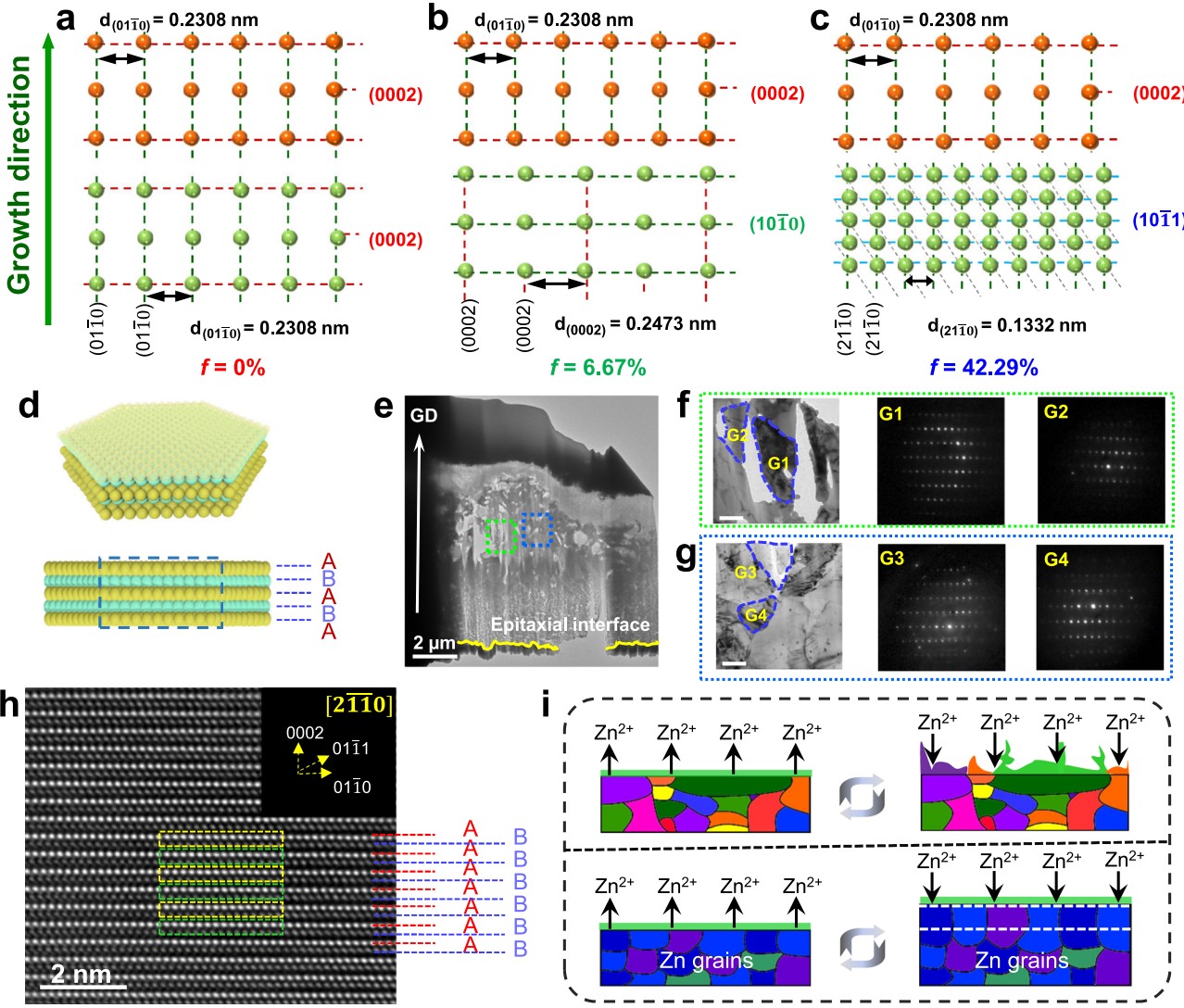

**Fig. 4 | The homoepitaxial mechanism of Zn(0002) electrodes. a–c** The lattice mismatch of Zn(0002), Zn(10$\bar{1}$0) and Zn(10$\bar{1}$1) to Zn(0002), respectively. **d** The structural illustration of the atomic arrangement of the hexagonal Zn nanosheet. **e** Typical cross-section TEM image of the deposited Zn on the Zn(0002) surface along growth direction. The box outlines the area magnified in **f**, **g**. **f**, **g** Magnified TEM image of the deposited Zn and the NBED patterns of the selected Zn grains (scale bar, 200 nm). **h** The high-resolution TEM image of the deposited Zn. The NBED pattern of component of deposited Zn (inset). **i** Schematic illustration of the Zn stripping/plating on the surface of com-Zn and Zn(0002) electrodes.

1 h under a current density of 4 mA cm$^{-2}$. In contrast, the stress of Zn(0002) metal electrode changed slightly from 14.8 MPa to 15.7 MPa, indicating negligible lattice strain and mismatch dislocations of deposited Zn. These findings suggest that while employing a Zn metal electrode with a predominant (0002) texture promotes uniform Zn deposition at the initial stage, the subsequently deposited Zn shows irregular and random growth once the deposited layer surpasses the critical thickness of homoepitaxial growth.

### Homoepitaxial mechanism of Zn on Zn(0002) metal electrode

A tolerance lattice mismatch ($f < 18\%$) between overgrowth crystal and substrate crystal is a key factor in achieving epitaxial growth; this small lattice mismatch is conducive to minimizing the interfacial energy of the two crystals[39]. However, it should be noted that the aforementioned theory is exclusively applicable to thermodynamically stable states[50]. In battery systems, particularly at high current densities, the Zn growth rate is considerably accelerated, resulting in a kinetic-dominated process. Furthermore, this theory is applicable to a substrate crystal with a single crystalline orientation (Supplementary Fig. 17a). The presence of even a small amount of other Zn textures in

the Zn(0002) metal electrode induces dislocation at the epitaxial interface (Supplementary Fig. 17b), causing lattice distortion. Beyond the critical thickness for epitaxial growth, subsequently deposited crystals display a disordered crystalline orientation, indicating the failure of the homoepitaxial mechanism. Therefore, it is imperative to ensure the Zn(0002) metal substrate with a single crystalline orientation to eliminate the lattice mismatch at epitaxial interface and provide uninterrupted driving forces for Zn epitaxial growth.

Based on a series of lattice mismatch calculations, it has been observed that a persistent lattice mismatch exists between the Zn(0002) plane and the Zn(10$\bar{1}$0) and Zn(10$\bar{1}$1) planes, with magnitudes of 6.67% and 42.29%, respectively, as illustrated in Fig. 4b, c. The electrodeposited Zn(0002) electrode presented a hexagonal close-packed (hcp) structure (Fig. 4a), in which the Zn(0002) planes were stacked parallelly in the order of "-ABABAB-" mode[51], displaying a single Zn(0002) texture, as depicted in Fig. 4d. To conclusively clarify the pivotal role of the substrate crystal featuring a single-Zn(0002) crystalline orientation in determining the growth direction of Zn, a nano-beam electron diffraction (NBED) technique was employed in transmission electron microscopy to investigate the underlying

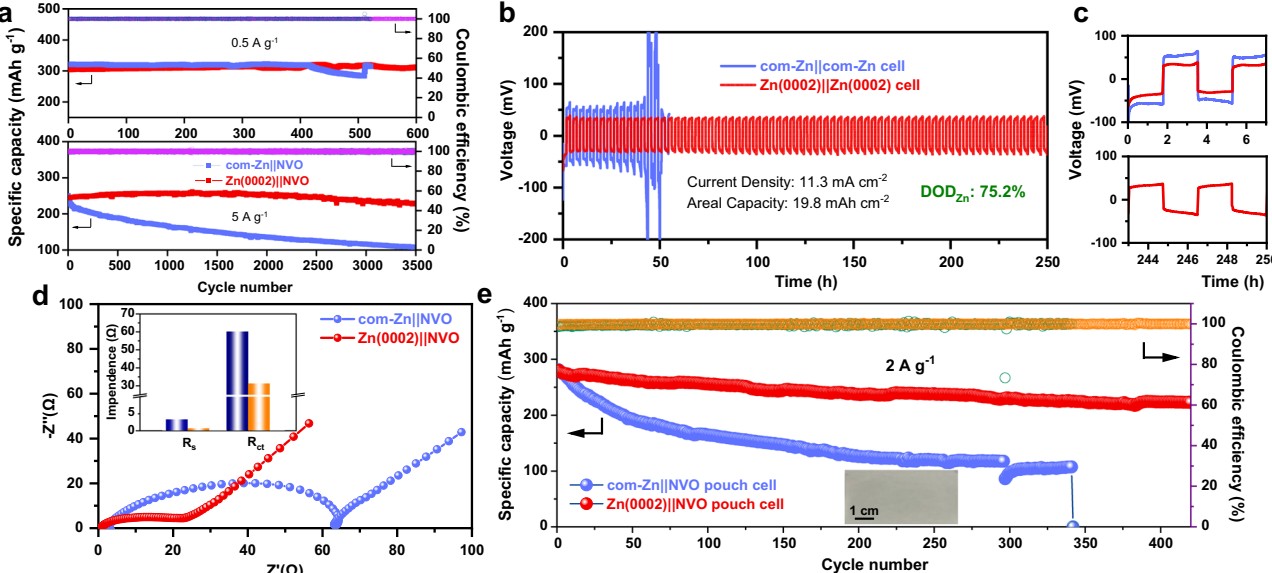

**Fig. 5 | Electrochemical performances and practical application of aqueous Zn|| NVO cells with the Zn(0002) metal anode. a** Cyclic capability of the Zn||NVO coin cells using com-Zn and Zn(0002) anodes at different current densities (0.5 A g⁻¹ and 5 A g⁻¹). **b** Cycling performance of the Zn||Zn cells with a limited Zn supply (DOD$_{Zn}$ = 75.2%), a current density of 11.3 mA cm⁻², and a areal capacity of 19.8 mAh cm⁻². **c** The selected enlarged voltage profiles of **b. d** Electrochemical impedance spectra curves of the com-Zn||NVO and the Zn(0002)||NVO cells. The inset shows related dependence of $R_s$ and $R_{ct}$. **e** Cycling performance of the Zn||NVO pouch cell using com-Zn and electrodeposited Zn(0002) anodes. The areal capacity of pouch cells is ~2 mAh cm⁻².

epitaxial mechanism in the Zn(0002) metal. In this regard, TEM foils of the metal were sliced from the deposited Zn to the epitaxial interface along the direction perpendicular to the epitaxial substrate via focused ion beam technology. We observed sharp interfaces between the Zn(0002) substrate and the Zn deposits from cross-sectional TEM observations (Fig. 4e). Several grains (G1, G2, G3, and G4) of the overgrowth crystal were randomly chosen to characterize its growth direction using electron diffraction analysis under TEM, as shown in Fig. 4f, g. The diffraction spots revealed that the lattice of the grains had the exact same lattice orientation. Zooming in on nanograins of the deposited Zn under high-resolution TEM, we observed that the Zn atoms derived from the Zn deposits were perfectly matched with the Zn(0002) substrate crystal by the order of "-ABABAB~" arrangement (Fig. 4h). As presented in the illustration of Fig. 4i, the such Zn(0002) metal was assumed to enable an epitaxial, rather than a dendritic, growth of Zn to sustain the consistency of the substrate texture. Consequently, Zn metals with a single (0002)-textured crystal-lographic feature enable ultra-sustainable homoepitaxial growth.

### Electrochemical performance of Zn metal electrodes

To accurately assess the reversibility of Zn metal electrodes, their CE must be quantified in Cu||Zn half cells. Cu||com-Zn cells exhibited failure after only 97 cycles at 1 mA cm⁻²/1 mAh cm⁻², whereas Cu|| Zn(0002) cells displayed an impressive average CE of 99.6% over 330 cycles (Supplementary Fig. 18a). Even at a large current density and capacity of 5 mA cm⁻² and 5 mAh cm⁻², the Zn(0002) electrode could stably cycle for 220 cycles, with a high average CE of 99.5% due to its superior corrosion resistance of Zn(0002) electrode (Supplementary Fig. 18b). The longevity and voltage polarization of Zn||Zn symmetric cells were further analyzed, where Zn||Zn cells comprising Zn(0002) electrodes demonstrated a longer cycle life and lower voltage polar-ization than those with com-Zn electrodes (Supplementary Fig. 19a–c). Furthermore, the rate performance of Zn||Zn cells with Zn(0002) electrodes surpassed those with com-Zn electrodes, confirming the high reversibility of the Zn(0002) metal electrode (Supplementary Fig. 20)[52]. A half-cell experiment was conducted on Zn electrodes with a 3-electrode configuration to calculate the hydrogen evolution

reaction (HER) rate by linear sweep voltammetry (LSV) curves[53,54]. It is evident that the HER of the Zn(0002) electrode occurred at a more negative potential than that of com-Zn electrode (Supplementary Fig. 21a, b). These results demonstrate that the single (0002)-textured Zn metal electrode can effectively inhibit HER in comparison with com-Zn, attributed to the superior thermodynamic stability of the Zn(0002) crystal plane. Additionally, XRD, X-ray photoelectron spec-troscopy (XPS) and electron probe microanalysis (EPMA) were employed to further investigate their corrosion behavior. As shown in Supplementary Fig. 22, the new peak on com-Zn at around 8.25°, identified to Zn₄SO₄(OH)₆·5H₂O (ZHS) by-products[55], was significantly reduced on Zn(0002), suggesting its corrosion resistance effect. This result was consistent with the XPS spectra and EPMA wavelength-dispersive X-ray spectroscopy (WDS) images of the cycled Zn elec-trode, as depicted in Supplementary Figs. 23a, b and 24a, b. Con-sidering the remarkable increase in the nucleation and growth rate of Zn at large currents, the stability of the Zn||Zn cells under 75.2% DOD$_{Zn}$ was tested using thin Zn metal electrodes (45 μm, 26.3 mAh cm⁻²). The results presented in Fig. 5b, c demonstrated that the corresponding Zn||Zn cells with Zn(0002) electrodes can stably cycle for over 250 h, while those with com-Zn electrodes were short-circuited after 51 h. Therefore, the Zn(0002) metal electrode has significant potential for practical applications to obtain improved performance and longer lifespans.

The Zn metal electrodes were also tested in full cells configuration using a NH₄V₄O₁₀ (NVO) cathode and 3 M ZnSO₄ electrolyte. The com-Zn||NVO cells exhibited rapid capacity decay, while the Zn(0002)||NVO cells delivered cycling stability of up to 600 and 3500 cycles with a capacity retention of up to 98% at 0.5 A g⁻¹ and 5 A g⁻¹, respectively (Fig. 5a). The normalized discharge/charge profiles of the Zn||NVO cells revealed that the com-Zn||NVO cell had a greater voltage hysteresis than the Zn(0002)||NVO cell (Supplementary Fig. 25). The initial cells of Zn||NVO were characterized by a Nyquist plot to enable the extra-polation of the charge transfer resistance ($R_{ct}$). The EIS data (Fig. 5d) indicated that the $R_{ct}$ of the Zn(0002)||NVO cell (31.35 Ω) was sub-stantially less than that of the com-Zn||NVO cell (60.37 Ω), suggesting that Zn(0002) anode facilitated charge transfer kinetics[56]. Even after

1000 cycles, the Zn(0002)||NVO cell still delivered a significantly reduced $R_{ct}$ than that of the com-Zn||NVO cell, indicating that the Zn(0002) held promise in regulating Zn epitaxial growth and suppressing side reactions (Supplementary Fig. 26). Besides, the lower $R_{ct}$ of Zn(0002) compared to that of com-Zn was attributed to its higher exchange current density ($i_0$) (Supplementary Fig. 27), indicating the fast deposition kinetic of the Zn(0002) electrode. The Zn(0002)|| NVO cell exhibited higher specific capacities than those with com-Zn anode at specific current of 1, 2, 5, 10, and 20 A g⁻¹ (Supplementary Fig. 28a–c). This result was supported by the cyclic voltammetry (CV) response of the full cell at a scan rate of 1 mV s⁻¹ (Supplementary Fig. 29). The SEM images of Supplementary Fig. 30 demonstrated that, in contrast to the com-Zn anode, the Zn(0002) grains remained fully intact even after 1000 cycles, strongly indicating the ultra-sustainable epitaxial growth of Zn(0002) metal during cycling. Cycling tests of Zn||NVO pouch cells were conducted in Fig. 5e, and the capacity retention of Zn(0002)||NVO pouch cells was 80% after 450 cycles but that of com-Zn||NVO pouch cells was only 61% after 100 cycles and showed unstable CEs. Furthermore, Zn(0002)||NVO pouch cells generated a stable current that illuminated a light and rotated a fan (Supplementary Fig. 31a, b; Supplementary Movie 1, 2).

## Discussion

In this study, we successfully electrodeposited [0001]-uniaxial oriented Zn(0002) metal anodes by regulating the anion species present in electrolyte. Furthermore, we utilized a nano-beam electron diffraction technique in TEM to comprehensively investigate the homoepitaxial growth mechanism of Zn on Zn(0002) metal surface. These results highlight the significance of maintaining only one crystalline orientation in Zn(0002) metal anode for sustainable homoepitaxial growth. Even at a high DOD$_{Zn}$ of 75.2%, the Zn(0002) symmetric cells exhibit a 250 h operational capability. Therefore, our investigation on homoepitaxial growth in the battery system is expected to stimulate further optimization of metal electrodes, which could lead to ultra-sustainable reversible cycling.

## Methods

### Preparation of Zn metal electrodes

Copper (Cu) foil (100 μm, 99.9999% metal basis, Alfa) was rinsed with 1% dilute sulfuric acid (H₂SO₄) to remove surface oxides and increase active sites before serving as electrodeposition substrate. Electrodeposited Zn metal electrodes with a thickness of about 127 μm were synthesized by using direct-current electrodeposition technique. A Cu sheet of with a diameter of 15 mm was used as cathode and a pure Zn plate as the anode. The distance between the anode and cathode was ~5 cm. For the Zn(0002) metal electrodes, the electrolyte contains 100 g/L ZnSO₄·6H₂O and 20 g/L boric acid (H₃BO₃). The electrolyte bath was stirred mechanically at 700 r/min. The deposition was performed 1 h at 25 °C with a current density of 30 mA cm⁻² and a pH value of 2. The IMS-Zn(0002) metal with a relative texture coefficients (RTCs) of 93 for the Zn(0002) lattice plane was synthesized by adding 5 g/L Zn(CH₃COO)₂·2H₂O in the above solution. For other Zn metal electrodes, the ZnSO₄·6H₂O was replaced by ZnCl₂ or Zn(CH₃COO)₂·2H₂O and other conditions remained the same.

### Preparation of NH₄V₄O₁₀ cathode

1.170 g NH₄VO₃ was dissolved in 80 °C deionized water to form a light-yellow solution. 1.891 g H₂C₂O₄ • 2H₂O solid powders were then added to the solution under magnetically stirring until it turned black-green. The resulting solution was transferred to a 50 mL autoclave and kept in an oven at 140 °C, 3 MPa for 48 h. Upon cooling to room temperature, the products were collected and washed repeatedly with deionized water, and subsequently dried at 60 °C for 12 h to obtain the NH₄V₄O₁₀ (NVO) materials.

## Materials characterizations

Structural characterization and texture types of Zn metal electrodes during stripping/plating were conducted on an X-ray diffractometer (XRD, Rigaku Mini Flex 600 diffractometer) operated at 100 mA and 40 kV with Cu-Kα radiation. The plane-view microstructures and grain sizes of the Zn metal electrodes were examined in a FEI Nova NanoSEM 430 field emission gun scanning electron microscope (FEG-SEM) with backscattering electron (BSE) imaging using a VCD detector. Electron backscatter diffraction (EBSD, NordlysMax2) measurements and analysis were performed on the plane view of the Zn metal electrodes using the HKL channel 5e software suite. A Tecnai G² F20 transmission electron microscope operated at 200 kV was used for TEM and HRTEM observations. Thin foils for SEM, EBSD, and TEM observations were prepared by removing the part of deposits from the substrates, mechanically grinding followed by electro-polishing in a solution of alcohol and perchloric acid at 0 °C. The plane-view microstructures of the Zn(0002) metal were characterized using a JEOL 2010 transmission electron microscope at an accelerating voltage of 200 kV.

## Growth mechanism characterizations

The crystalline orientation and Zn atoms arrangement of deposited Zn on the Zn(0002) metal surface were characterized using high angle angular dark filed scanning transmission electron microscopy (HAADF-STEM). The Zn(0002) metals were first plated 1 h and then cross-sectioned parallel to the growth direction (GD) using a focused ion beam (FIB), and only those areas close to the epitaxial interface were punched out for observation.

## Electrochemical measurements

The electrochemical performance of the batteries was evaluated using CR2025 coin-type cells on a multichannel battery test system (LAND CT2001A, China) at room temperature of 25 °C. Glass fiber (Whatman GF/C, 260 μm) filter paper was utilized as the separator. All the electrochemical tests are run at 25 °C. CE measurements were conducted using asymmetric Zn||Cu cells. Tafel curves measurements in 3 M ZnSO₄ electrolyte were carried out on an electrochemical workstation (Multi Autolab/M204) using a three-electrode system with com-Zn or Zn(0002) electrodes as the working electrode, platinum (Pt) foil as the counter electrode and Ag/AgCl as the reference electrode. The cathode was composed of NVO (1.2–1.5 mg cm⁻²), conductive carbon black and PVDF at a mass ratio of 7:2:1. The mass of cathode, separator, and Zn metal anode are 24.24, 34.72, and 96.18 mg, respectively. The Zn||NVO full cells with 3 M ZnSO₄ electrolyte (100 μL) were tested in the voltage range of 0.4–1.4 V. Current densities were conducted in Zn||Zn symmetric cells and Zn||Cu asymmetric cells are calculated based on the area of the Zn or Cu electrodes. Specific capacities and current densities in the Zn||NVO full cells are calculated based on the mass of active material in the cathode. Linear sweep voltammetry (LSV), CV and electrochemical impedance spectroscopy (EIS) tests were performed on an electrochemical workstation (CHI660E, China). DOD of the Zn anode is calculated by dividing the cycled capacity into the total theoretical capacity of the Zn anode. The pouch cell (30 × 30 mm) was tested under specific pressure conditions using the multichannel battery tester (Neware CT-4008T) within a climatic chamber (BOLING BLC-300) matained at 25 °C.

## Theoretical calculations

The density functional theory (DFT) calculations had been performed by the Vienna Ab initio Simulation Package (VASP) of the Perdew-Burke-Ernzerhof (PBE) formulation in the generalized gradient approximation (GGA). Based on the projected augmented wave (PAW) potential, the kinetic energy cut-off was set to 520 eV for the plane wave basis set, which was enough to describe the valence electrons. For the Brillouin zone sampling, we chose the k-meshes grides to 0.03*2π/Å. We employed the DFT-D3 method to describe the behavior

of the van der Waals interactions. The Gaussian smearing method with a width of 0.05 eV was used to capture the partial occupancies of the Kohn–Sham orbitals. For the geometry optimization, the convergence of force on each atom was set to be smaller than 0.01 eV/Å. For accurate energy calculations, the convergence for electronic energy was set to be smaller than $10^{-7}$ eV.

To obtain the surface energy of Zn crystal planes, we used the slab model with a 15 Å vacuum to avoid the interaction between surface atoms. The thickness of the atomic layers of different surfaces was more than 12 Å to prevent surface interactions. The VASPsol method of the solvent model was used to simulate the aqueous solution for the series of surface calculations including the surface energy and adsorption behaviors. The surface energy ($\sigma$) can be defined as the following: $\sigma = (E_s - n\mu_{Zn})/2A$, where $E_s$ is the energy of the slab model, $\mu_{Zn}$ is the chemical potential of the Zn atom in the bulk structure, $n$ is the number of Zn atoms in the slab model, $A$ is the surface area. We used 5, 7, 9, and 11 atomic layers of the slab models for fitting to obtain reasonable surface energies. Here, the above equation can be modified as: $E_s = 2A \times \sigma + n\mu_{Zn}$. The surface energy can be fitted from the relations between $E_s$ and $n$. The surface energy can be influenced by the adsorption of ions or molecules and the adsorption energy can be written as: $E_{ads} = E_t - E_s - n\mu_i$, where $E_t$ is the total energy of the adsorption model and $E_s$ is the energy of the slab model, $\mu_i$ is the chemical potential for the adsorbed ions or molecules. The modified surface energy $\sigma^*$ after adsorption can be defined as follows: $\sigma^* = \sigma + E_{ads}/A$. Here we used the growth theory proposed by Wulff to determine the configuration of nanoparticles. Based on the crystal spatial symmetry, the Wulff configuration can be calculated by different surface energies, and the coverage $\theta$ of different surfaces can also be obtained.

## Reporting summary

Further information on research design is available in the Nature Portfolio Reporting Summary linked to this article.

## Data availability

All data that support the findings of this study are presented in the Manuscript and Supplementary Information, or are available from the corresponding author upon request. Source data are provided with this paper.

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

## Acknowledgements

The research was supported by grants from the National Natural Science Foundation of China (52202339 (X.Z.), 52372252 (J.Z.), 51932011 (S.L.)). Meanwhile, we are grateful for resources from the High-Performance Computing Center of Central South University.

## Author contributions

All authors contributed to this work. X.Z., J.Z. and S.L. conceived the concept for the research. X.Z. designed the experiments and analyzed data with assistance from Y.L. and B.L.. J.X. conducted simulate calculations and helped to write this part. X.Z. and J.Z. discussed together, wrote the manuscript and made extensive revisions to the original manuscript.

## Competing interests

The authors declare no competing interests.
