## [Peer Review File · Nature Communications]

Single [0001]-oriented zinc metal anode enables sustainable zinc batteriesReviewers' comments:

Reviewer #1 (Remarks to the Author):

This manuscript reports an analysis of the crystallographic effect of the homoepitaxial Zn(0002) surface on reversible Zn anode cyclability. The authors demonstrated successful formulation of (0002) plane-rich Zn film growth on Cu substrate by electrodeposition in specific electrolytes, including H₃BO₃ and sulfate anion. Structural characterizations reveal large uniformity of the deposited Zn film in crystallographic orientation (i.e., (0002) plane) compared to hetero-structured commercial Zn foil, which leads to improved structural uniformity and long-term cycling performance of Zn anode. Therefore, this manuscript offers an interesting electrodeposition approach to formulate beneficial Zn(0002) plane selectively on a polycrystalline substrate like Cu foil. However, several points need to be addressed more thoroughly to enhance clarity and improve the quality of manuscript to be suitable publication in Nature Communications.

1. The authors have used electrodeposition to obtain homoepitaxially (0002) plane-rich Zn film on Cu. From an electrodeposition standpoint, this result is intriguing as electrolyte components plays a crucial role in determining the crystallography of the top-surface during electrodeposition. While the author provided thermodynamic rationales of surface energy differences depending on different anion types, however, more detailed descriptions are necessary for a more thorough understanding of the underlying mechanism during the electrodeposition. Therefore, I strongly recommend expanding the discussions to include the role of H₃BO₃ in the structural quality of deposited Zn film, supported by additional experimental results.

2. Building upon the previous comment, it is widely known that the quality of the electrodeposited films is highly dependent on various experimental conditions, including current density, mass transfer condition, and electrolyte concentration. Consequently, even with the same electrolyte composition, the quality of the Zn(0002) plane (e.g. crystal domain size) can be significantly affected by those parameters, which will influence the overall sustainability of the Zn anode at different levels. Thus, it is recommended to provide additional insights into the correlation between the electrodeposition conditions mentioned above and the homoepitaxial quality of the Zn(0002) plane on the deposited film.

3. While the authors primarily focuses on the structural evolution of the Zn anode during charge/discharge cycles on the anode side, it would be valuable to investigate hydrogen evolution reaction (HER) rate on the Zn surface, which is the other critical issue for sustainable Zn-ion battery. The author should include a comparison of HER rate and consequent Coulombic efficiency of the cell. Ideally, conducting a half-cell experiment on the Zn anode with a 3-electrode configuration is recommended for more fundamental understanding and insightful discussions.

4. In Figure 3, the change of (0002) plane in X-ray diffraction (XRD) analysis is not distinguishable enough. It is recommended to modify either color of the (0002) signal or its visualization.

5. The (0002) plane-rich Zn film showed lower charge transfer resistance than the commercial Zn foil in the Nyquist plot. However, in-depth discussion to rationalize this result is limited to provide clear understanding. The authors should elaborate the origin of this lower resistance in the (0002) plane-rich Zn surface compared to hetero-structured commercial one, specifically focusing on the reactive Zn anode.

6. The authors argued that structural homogeneity in (0002) plane results in a lower formation of Zn₄SO₄(OH)₆·5H₂O by-products. Indeed, presence of byproducts or chemical state of Zn surface over repeated cycling is critical issue to analyze the origin of Zn anode stability. Hence, it is highly recommended to provide X-ray photoelectron spectroscopy (XPS) data of the cycled Zn anodes.

Reviewer #2 (Remarks to the Author):

This manuscript reports the utilization of electrodeposited [0001]-uniaxial oriented Zn(0002) metal

anodes in aqueous zinc-ion batteries. The authors claim that this Zn(0002) metal anode exhibits exceptional epitaxial growth, even under challenging conditions such as high depth of discharge and/or high areal capacity. While the authors have made efforts to enhance the electrochemical performance and have conducted various analyses to validate their claim, previous report has already explored the use of a single crystal Zn metal anode, and the evidence provided to support the author's claim is unreliable and often confusing. Therefore, considering the esteemed reputation of this journal, I do not support the publication of this manuscript in *Nature Communications*. Please refer to the comments below.

1. The most concerning issue of this work is its lack of novelty. A previous report has already demonstrated that a single crystal (002) Zn metal anode in ZnSO₄ aqueous electrolyte exhibits remarkably high electrochemical performance compared to the use of other substrate materials, including polycrystalline Zn [*Adv. Mater.* **2022**, *34*, 2202552].
2. The authors employed an H₃BO₃/ZnSO₄-based solution to fabricate the Zn(0002) metal anode. However, upon examining the XRD results from the reference paper [Ref. 31], it becomes evident that despite Zn being electrodeposited in an electrolyte with the same components, additional crystallographic orientations coexist with (0002). In comparison to the previous paper, how does this manuscript account for the exclusive growth of Zn in the (0002) direction? Considering that the reference paper also utilized SO₄²⁻ anion in its electrolyte but displayed orientations other than (0002), it appears that factors such as stirring or high current density, which differ from the conditions in that paper, exert a more dominant influence on determining the preferred deposition orientation of Zn, surpassing the impact of the anion type.
3. The authors used Cu as the substrate to prepare the Zn(0002) metal anode. Despite a significant lattice mismatch between the Cu substrate and the subsequently electrodeposited Zn, the (0002) directing of Zn was still achievable. This suggests that the composition of the electrolyte bath or the operating conditions of the electrodeposition process, independent of lattice mismatch, act as dominant factors in regulating the texture through different mechanisms. Therefore, there appears to be a contradiction between the claim of lattice mismatch in this paper and the observed phenomenon.
4. The authors claim anion-induced texture regulation of Zn crystals; however, the comparative results they provided based on different anions only include calculations of surface energy and XRD data of the electrode. However, there is a possibility that substances like boric acid present in the electrolyte bath during the preparation of the Zn(0002) metal anode could have adsorbed onto the surface or caused surface modifications such as SEI formation. Therefore, to truly validate the effects resulting from differences in the deposition crystallographic orientation due to anions, it would be necessary to present electrochemical data using electrodes deposited under the same conditions, except for using different anions as a control group, rather than relying solely on commercial Zn. This would provide substantial evidence to substantiate the influence of anions on the deposition orientation, beyond mere speculation.

Response to Reviewers' Comments

We sincerely appreciate the invaluable comments and insightful suggestions provided by the reviewers for our manuscript. After careful consideration, we conducted additional experiments and made comprehensive revisions to improve the manuscript's overall content and central focus. Below, you will find a detailed description of the implemented changes, accompanied by our point-by-point response to the reviewers' comments.

Reviewer #1

This manuscript reports an analysis of the crystallographic effect of the homoepitaxial Zn(0002) surface on reversible Zn anode cyclability. The authors demonstrated successful formulation of (0002) plane-rich Zn film growth on Cu substrate by electrodeposition in specific electrolytes, including H_3BO_3 and sulfate anion. Structural characterizations reveal large uniformity of the deposited Zn film in crystallographic orientation (i.e., (0002) plane) compared to hetero-structured commercial Zn foil, which leads to improved structural uniformity and long-term cycling performance of Zn anode. Therefore, this manuscript offers an interesting electrodeposition approach to formulate beneficial Zn(0002) plane selectively on a polycrystalline substrate like Cu foil. However, several points need to be addressed more thoroughly to enhance clarity and improve the quality of manuscript to be suitable publication in Nature Communications.

Response:

We appreciate the reviewer's acknowledgement of the major achievements of our study. The constructive feedback provided by the reviewer contributes significantly to a deeper understanding of our findings and the underlying mechanisms of this research. This, in turn, enhances the overall quality of our manuscript. Firstly, we apologize for the lack of specificity in describing the preparation process of the single Zn(0002)-textured electrodes in the initial manuscript, resulting from our inadequate and deficient descriptions. In response to the reviewers' feedback, we have provided a more detailed and comprehensive elaboration on the essential requirements for creating a 100% (0002)-textured Zn metal electrode. This involves systematic modification of various variables within the electroplating preparation process, including the electrolyte components, current density and pH value. Secondly, we notice that the reviewer expressed more concern regarding the preparation techniques and parameter

control in the electrodeposition process, rather than emphasizing the importance of Zn metal electrodes possessing a single Zn(0002) texture for epitaxial sustainability. We recognize that this crucial aspect was not adequately highlighted in our initial manuscript. Consequently, we conducted additional experiments by preparing Zn metal electrodes with a relative texture coefficients (RTCs) of 93 for the Zn(0002) lattice plane as intermediate states (designated as IMS-Zn(0002)) to further elucidate the central focus of our manuscript. By addressing these issues and incorporating the reviewer's feedback, we believe that our manuscript is now significantly improved. We sincerely hope that with these revisions, you will reconsider the publication of this work in *Nature Communications*.

1. The authors have used electrodeposition to obtain homoepitaxially (0002) plane-rich Zn film on Cu. From an electrodeposition standpoint, this result is intriguing as electrolyte components plays a crucial role in determining the crystallography of the top-surface during electrodeposition. While the author provided thermodynamic rationales of surface energy differences depending on different anion types, however, more detailed descriptions are necessary for a more thorough understanding of the underlying mechanism during the electrodeposition. Therefore, I strongly recommend expanding the discussions to include the role of H₃BO₃ in the structural quality of deposited Zn film, supported by additional experimental results.

Response:

We express our gratitude to the reviewer for highlighting the role of H₃BO₃ in determining the crystalline orientation of Zn electrodes during the electrodeposition process. This insight has enabled us to provide comprehensive and supplementary information regarding the electrodeposition parameters. The single (0002)-textured Zn metal electrode was electrodeposited on Cu foil under a current density of 30 mA cm⁻² in solutions containing 100 g L⁻¹ ZnSO₄·6H₂O and 20 g L⁻¹ H₃BO₃, with a pH value of 2 and vigorous agitation (700 r min⁻¹). It is important to note that the vigorous agitation during the deposition process serves to eliminate H₂ generated during the Zn plating process, preventing gas accumulation on the substrate and enhancing deposit quality (*Surface & Coating Technology*, 2006, 201, 371-383; *Surface & Coating Technology*, 2002, 157, 282-289). Moreover, intense stirring guarantees that Zn deposition remains unaffected by the substrate, as it minimizes the contact duration between the zinc ions and the substrate. Boric acid (H₃BO₃), as mentioned by the reviewer, is a commonly used buffer agent in acidic deposition solutions, primarily serving to stabilize

the pH value (*Journal of Pharmaceutical Science*, 2020, 2375-2386; *Journal of the Electrochemical Society*, 2020, 167, 11250). As described in the specialized electroplating manual by Kanani N. (*Electroplating: basic principles, processes and practice. Elsevier. 2004*), the presence of H_3BO_3 ensures the relative stability of the pH value in the plating solution. This is attributed to H_3BO_3 being a triprotic weak acid, capable of dissociating or absorbing hydrogen ions in response to fluctuations in the plating solution's pH ($H_3BO_3 \leftrightarrow H_2BO_3^- + H^+$; $H^+ + OH^- \leftrightarrow H_2O$).

To attain a more thorough comprehension of the impact of various electroplating parameters, including H_3BO_3 additive, pH value, and current density, on the metal texture of electrodeposited Zn electrodes, a series of experiments were conducted (**Supplementary Table 1**), as detailed below:

i) Under 30 mA cm^{-2} current density and vigorous stirring, the electrodeposited Zn metal electrodes in 100 g L^{-1} $ZnSO_4$ solution led to the formation of $Zn_4SO_4(OH)_6 \cdot 5H_2O$ (ZHS) by-products on its surface due to severe side reactions, resulting in a disordered Zn texture (**Supplementary Fig. 1a, Supplementary Fig. 2a and Supplementary Fig. 3a**).

ii) Under 30 mA cm^{-2} current density and vigorous stirring, the electrodeposited Zn metal electrodes in 100 g L^{-1} $ZnSO_4$ solution with $\text{pH} = 2$ can maintain a single crystalline orientation in the initial 10 min. However, with an increase in deposition time, the deposition solution experienced a decrease in pH value, triggering enhanced side reactions and a reduction in deposition efficiency. Consequently, this phenomenon resulted in the formation of a disordered texture and an uneven surface on the prepared Zn electrodes (**Supplementary Fig. 1b, Supplementary Fig. 2b and Supplementary Fig. 3b**).

iii) Under 30 mA cm^{-2} current density and vigorous stirring, the electrodeposited Zn metal electrodes in 100 g L^{-1} $ZnSO_4$ and 20 g L^{-1} H_3BO_3 solution with $\text{pH} = 2$ presented a single Zn(0002) texture without any by-products (**Supplementary Fig. 1c, Supplementary Fig. 2c and Supplementary Fig. 3c**). This result can be attributed to the incorporation of H_3BO_3 and vigorous stirring. These factors collectively create an ideal environment, enabling maximum exposure of the Zn(0002) crystal plane with the lowest surface energy, while remaining unaffected by the substrate.

iv) Regulating the current density within the parameters of Experiment iii revealed intriguing findings. At lower currents (10 and 20 mA cm^{-2}), it became feasible to fabricate zinc metal electrodes with a single (0002) texture (**Supplementary Fig. 3d**). However, this achievement was accompanied by reduced deposition efficiency and surface unevenness (**Supplementary Fig. 1d and**

Supplementary Fig. 2d). Conversely, higher current density (50 mA cm^{-2}) exacerbated side reactions and uncontrolled Zn growth, leading to a disordered Zn texture (**Supplementary Fig. 2d and Supplementary Fig. 3d**). Consequently, a current density of 30 mA cm^{-2} appears to be the optimal electroplating condition.

In summary, stirring, H_3BO_3 additives, pH values, and appropriate current density are essential factors in creating an optimal electroplating environment for fabricating a single (0002)-textured Zn metal electrode. Therefore, in this ideal environment, within the ZnSO_4 system, as opposed to the ZnCl_2 and $\text{Zn}(\text{CH}_3\text{COO})_2$ systems, the surface energy of the Zn(0002) crystal plane undergoes the most significant reduction (**Fig. 1e,f**), making it easier to expose.

Changes:

This was achieved through the incorporation of H_3BO_3 additive and vigorous stirring, creating an ideal environment to eliminate harmful side reactions³¹⁻³⁵. As a result, the crystallographic orientation and texture of the electrodeposited Zn metals were exclusively dependent on the anions present in the deposited liquids. Detailed experimental parameter regulation is provided in **Supplementary Table 1** and **Supplementary Fig. 1-3**. The Zn metals deposited from ZnCl_2 and $\text{Zn}(\text{CH}_3\text{COO})_2$ systems displayed irregular crystalline orientation, whereas a single (0002)-textured Zn metal was obtained from the ZnSO_4 system, as shown in the scanning electron microscopy (SEM) images (**Fig. 1b-d**). (In the Manuscript, Page 4-5)

Supplementary Table 1. The electrodeposition parameters of Zn metal electrodes

Experiment number	i	ii	iii	iv
Electrodeposition parameters	100 g L^{-1} $\text{ZnSO}_4 \cdot 6\text{H}_2\text{O}$, $J = 30 \text{ mA cm}^{-2}$	100 g L^{-1} $\text{ZnSO}_4 \cdot 6\text{H}_2\text{O}$, pH $=2$, $J = 30 \text{ mA cm}^{-2}$	100 g L^{-1} $\text{ZnSO}_4 \cdot 6\text{H}_2\text{O}$, pH =2, $20 \text{ g L}^{-1} \text{H}_3\text{BO}_3$, $J =$ 30 mA cm^{-2}	100 g L^{-1} $\text{ZnSO}_4 \cdot 6\text{H}_2\text{O}$, pH =2, $20 \text{ g L}^{-1} \text{H}_3\text{BO}_3$, $J =$ $10\text{-}50 \text{ mA cm}^{-2}$

Supplementary Fig. 1. Optical images of prepared Zn metal electrodes under the corresponding electrodeposition parameters of Supplementary Table 1. Experiment (a) i, (b) ii, (c) iii and (d) iv.

Supplementary Fig. 2. SEM images of prepared Zn metal electrodes under the corresponding electrodeposition parameters of Supplementary Table 1. Experiment (a) i, (b) ii, (c) iii and (d) iv.

Supplementary Fig. 3. XRD patterns of prepared Zn metal electrodes under the corresponding electrodeposition parameters of Supplementary Table 1. Experiment (a) i, (b) ii, (c) iii and (d) iv.

To attain a more thorough comprehension of the impact of various electroplating parameters, including H_3BO_3 additive, pH value, and current density, on the metal texture of electrodeposited Zn electrodes, a series of experiments were conducted (Supplementary Table 1), as detailed below:

i) Under 30 mA cm^{-2} current density and vigorous stirring, the electrodeposited Zn metal electrodes in $100 \text{ g L}^{-1} \text{ ZnSO}_4$ solution led to the formation of $\text{Zn}_4\text{SO}_4(\text{OH})_6 \cdot 5\text{H}_2\text{O}$ (ZHS) by-products on its surface due to severe side reactions, resulting in a disordered Zn texture (Supplementary Fig. 1a, Supplementary Fig. 2a and Supplementary Fig. 3a).

ii) Under 30 mA cm^{-2} current density and vigorous stirring, the electrodeposited Zn metal electrodes in $100 \text{ g L}^{-1} \text{ ZnSO}_4$ solution with $\text{pH} = 2$ can maintain a single crystalline orientation in the initial 10 min. However, with an increase in deposition time, the deposition solution experienced a decrease in pH value, triggering enhanced side reactions and a reduction in deposition efficiency.

Consequently, this phenomenon resulted in the formation of a disordered texture and an uneven surface on the prepared Zn electrodes (**Supplementary Fig. 1b, Supplementary Fig. 2b and Supplementary Fig. 3b**).

iii) Under 30 mA cm^{-2} current density and vigorous stirring, the electrodeposited Zn metal electrodes in $100 \text{ g L}^{-1} \text{ ZnSO}_4$ and $20 \text{ g L}^{-1} \text{ H}_3\text{BO}_3$ solution with $\text{pH} = 2$ presented a single Zn(0002) texture without any by-products (**Supplementary Fig. 1c, Supplementary Fig. 2c and Supplementary Fig. 3c**). This result can be attributed to the incorporation of H_3BO_3 and vigorous stirring. These factors collectively create an ideal environment, enabling maximum exposure of the Zn(0002) crystal plane with the lowest surface energy, while remaining unaffected by the substrate.

iv) Regulating the current density within the parameters of Experiment iii revealed intriguing findings. At lower currents (10 and 20 mA cm^{-2}), it became feasible to fabricate zinc metal electrodes with a single (0002) texture (**Supplementary Fig. 3d**). However, this achievement was accompanied by reduced deposition efficiency and surface unevenness (**Supplementary Fig. 1d and Supplementary Fig. 2d**). Conversely, higher current density (50 mA cm^{-2}) exacerbated side reactions and uncontrolled Zn growth, leading to a disordered Zn texture (**Supplementary Fig. 2d and Supplementary Fig. 3d**). Consequently, a current density of 30 mA cm^{-2} appears to be the optimal electroplating condition.

In summary, stirring, H_3BO_3 additives, pH values, and appropriate current density are fundamental factors in establishing an optimal electroplating environment to fabricate a single (0002)-textured Zn metal electrode. (In the Supporting Information, Page R2-R5)

2. Building upon the previous comment, it is widely known that the quality of the electrodeposited films is highly dependent on various experimental conditions, including current density, mass transfer condition, and electrolyte concentration. Consequently, even with the same electrolyte composition, the quality of the Zn(0002) plane (e.g. crystal domain size) can be significantly affected by those parameters, which will influence the overall sustainability of the Zn anode at different levels. Thus, it

is recommended to provide additional insights into the correlation between the electrodeposition conditions mentioned above and the homoepitaxial quality of the Zn(0002) plane on the deposited film.

Response:

Thank you for the reviewer's reminder. Indeed, as indicated in **Supplementary Table 1**, the quality of the electrodeposited Zn film is significantly dependent on the experimental conditions, including current density, mass transfer condition, and electrolyte concentration. It is noted that the Zn metal electrodes used in the battery system has a single (0002) texture, displaying higher corrosion resistance than other textured Zn metal electrodes due to the superior thermodynamic stability of Zn(0002) crystal plane (*Chemical Reviews*, 2020, 120, 7795-7866; *Advanced Materials*, 2021, 33, 2105951) and are not prone to induce the occurrence of corrosion and other side reactions. Therefore, the Zn plating process on the single (0002)-textured Zn metal electrode is different with the electrodeposited preparation process.

Furthermore, we conducted a series of experiments to elucidate the correlation between these electrodeposition conditions and the Zn-Zn homoepitaxial sustainability on the single (0002)-textured Zn metal electrodes in Zn symmetric cells. Firstly, our study focused on modulating the electrolyte's concentration, with ZnSO₄ electrolytes of 1 M, 2 M, and 3 M, respectively. As shown in **Fig. R1**, electrolyte concentration only affects the quantity and uniformity of deposited Zn on the Zn(0002) electrode, without disrupting Zn-Zn homoepitaxy continuity. Elevated concentrations contribute to a more uniform and flat Zn deposition due to sufficient zinc ions. Secondly, similar to the electrolyte concentration, the crystalline orientation of Zn deposition is exclusively dictated by the orientation of the Zn electrode substrate, regardless of the magnitude of the current density. On the one hand, the Zn symmetric/asymmetric cells and Zn full cells were assembled to perform a series of electrochemical and cycling tests under different current density (**Fig. 5** and **Supplementary Fig. 18, 19, 24, 27, 28**), in which the Zn(0002) metal electrodes sustain single (0002) textures. On the other hand, we further conducted deposition experiments on Zn(0002) metal electrodes using Zn symmetric cells under 0.5, 1, 2, 4, 8 mA cm⁻², respectively. Regardless of the magnitude of the current density, the Zn(0002) electrode within the Zn||Zn cells can maintain a consistent single crystalline orientation, unaffected by variations in current density (**Fig. R2**). However, with an increase in current density, there is a significant rise in the amount of Zn deposition on the electrode surface.

Figure R1. The effect of electrolyte concentration on Zn deposition behavior in Zn symmetric cells.

(a) 1 M, (b) 2 M, (c) 3 M ZnSO₄ electrolyte. (d) The corresponding XRD patterns of (a-c).

Figure R2. The effect of current density on Zn deposition behavior in Zn symmetric cells. Current

density, J = (a) 0.5, (b) 1.0, (c) 2.0, (d) 3.0, (e) 4.0, and (f) 8.0 mA cm⁻², respectively.

3. While the authors primarily focus on the structural evolution of the Zn anode during charge/discharge cycles on the anode side, it would be valuable to investigate hydrogen evolution reaction (HER) rate on the Zn surface, which is the other critical issue for sustainable Zn-ion battery. The author should include a comparison of HER rate and consequent Coulombic efficiency of the cell.

Ideally, conducting a half-cell experiment on the Zn anode with a 3-electrode configuration is recommended for more fundamental understanding and insightful discussions.

Response:

We thank the reviewer for carefully reviewing our manuscript and raising this point. In response to your valuable suggestions, we conducted a half-cell experiment on the Zn electrode with a 3-electrode configuration to analyze the HER rate by linear sweep voltammetry (LSV) curves. The potential of HER was tested in 1 M Na₂SO₄ aqueous solutions to avoid the interference from Zn reduction. It is evident that the HER of Zn(0002) electrode occurs at a more negative potential than that of com-Zn electrode (**Supplementary Fig. 20a**). Furthermore, owing to the constant generation of H₂ by water consumption, the cell based on com-Zn expands and detaches, causing electrolyte leakage and early battery failure (**Supplementary Fig. 20b**). These results demonstrate that the single (0002)-textured Zn metal electrode can effectively inhibit HER in comparison with com-Zn, which is attributed to the superior thermodynamic stability of the Zn(0002) crystal plane.

In general, the Coulombic efficiency (CE) of the Zn cells was tested in Zn||Cu cells. At 1 mA cm⁻²/1 mAh cm⁻², com-Zn cells failed after only 97 cycles, whereas Cu||Zn(0002) cells displayed an impressive average CE of 99.6% over 330 cycles due to the superior corrosion resistance of Zn(0002) electrode (**Supplementary Fig. 17a**). To further assess the reversibility of Zn metal electrodes, the CE was quantified in Zn||Cu cells at a relatively large current density and capacity of 5 mA cm⁻²/5 mAh cm⁻². Similarly, the com-Zn||Cu cell operated only 100 cycles, while the cell with Zn(0002) metal electrode could stably cycle for 220 cycles (**Supplementary Fig. 17b**). These results demonstrate that the single (0002)-textured Zn metal electrode with high reversibility effectively suppresses HER.

Changes:

A half-cell experiment was conducted on Zn electrodes with a 3-electrode configuration to calculate the hydrogen evolution reaction (HER) rate by linear sweep voltammetry (LSV) curves^{52,53}. It is evident that the HER of the Zn(0002) electrode occurred at a more negative potential than that of com-Zn electrode (**Supplementary Fig. 20a,b**). These results demonstrate that the single (0002)-textured Zn metal electrode can effectively inhibit HER in comparison with com-Zn, attributed to the superior thermodynamic stability of the Zn(0002) crystal plane. (In the Manuscript, Page 11)

Even at a large current density and capacity of 5 mA cm⁻² and 5 mAh cm⁻², the Zn(0002) electrode could stably cycle for 220 cycles, with a high average CE of 99.5% due to the superior corrosion

resistance of Zn(0002) electrode (**Supplementary Fig. 17b**). (In the Manuscript, Page 10)

The potential of HER was tested in 1 M Na₂SO₄ aqueous solutions in order to avoid the interference of Zn reduction. Furthermore, owing to the constant generation of H₂ by water consumption, the cell based on com-Zn expands and detaches, causing electrolyte leakage and early battery failure. (In the Supporting Information, Page R14)

Supplementary Fig. 20 | (a) LSV curves of the com-Zn, and Zn(0002) electrodes in 1 M Na₂SO₄ solution at a scan rate of 5 mV s⁻¹. (b) Optical images of Zn||Zn symmetric cells after plating/stripping at 2 mA cm⁻²/5 mAh cm⁻² for 200 cycles.

Supplementary Fig. 17 | Coulombic efficiencies of Zn plating/stripping process in com-Zn||Cu and Zn(0002)||Cu cells at (a) 1 mA cm⁻²/1 mAh cm⁻² and (b) 5 mA cm⁻²/5 mAh cm⁻².

4. In Figure 3, the change of (0002) plane in X-ray diffraction (XRD) analysis is not distinguishable

enough. It is recommended to modify either color of the (0002) signal or its visualization.

Response:

Thank you for the reviewer's reminder. To better distinguish the ratio changes of the Zn(0002) texture, we have included the original XRD comparison charts of Zn electrodes during the plating process in the Supplementary file, based on Fig. 3, for a clearer analysis. As shown in **Supplementary Fig. 16**, throughout the Zn deposition process, the intensity of the (002) peak in the X-ray diffraction (XRD) spectra gradually decreased for com-Zn metal electrodes. The XRD patterns showed that the deposited Zn on com-Zn and IMS-Zn(0002) surfaces lacked a fixed orientation, while a single (002) peak was consistently observed using the Zn(0002) electrode from the beginning to the end, illustrating ultra-sustainable epitaxial growth between the overgrowth and the substrate crystals. Furthermore, the (002) peak intensity of the deposited Zn using the Zn(0002) electrode was stronger, further demonstrating their perfect lattice match.

Changes:

Ex situ XRD was conducted to detect the structural evolution of Zn metal electrodes during the charging process, as shown in **Fig. 3b,d,f and Supplementary Fig. 16a-c**. The peaks located at approximately 36° in both electrodes were assigned to the (0002) plane of Zn crystal⁴⁸. The XRD patterns showed that the deposited Zn on com-Zn and IMS-Zn(0002) surfaces lacked a fixed orientation, while a single (002) peak was consistently observed using the Zn(0002) electrode from the beginning to the end, illustrating ultra-sustainable epitaxial growth between the overgrowth and the substrate crystals. Furthermore, the (002) peak intensity of the deposited Zn using the Zn(0002) electrode was stronger, further demonstrating their perfect lattice match. These findings suggest that while employing a Zn metal electrode with a predominant (0002) texture promotes uniform Zn deposition at the initial stage, the subsequently deposited Zn shows irregular and random growth once the deposition thickness surpasses the t_c of epitaxial growth. (In the Manuscript, Page 8)

Supplementary Fig. 16 | XRD patterns of Zn electrodes after plating 0, 5, 10, 20, 30 min, respectively. (a) com-Zn, (b) IMS-Zn(0002) and (c) Zn(0002).

5. The (0002) plane-rich Zn film showed lower charge transfer resistance than the commercial Zn foil in the Nyquist plot. However, in-depth discussion to rationalize this result is limited to provide clear understanding. The authors should elaborate the origin of this lower resistance in the (0002) plane-rich Zn surface compared to hetero-structured commercial one, specifically focusing on the reactive Zn anode.

Response:

Thanks for the reviewer’s constructive comment. According to the Butler-Volmer equation, the expression of the charge-transfer resistance (R_{ct}) changes into:

$$R_{ct} = \frac{RT}{nFi_0} \quad (1)$$

Where R is gas constant, T is temperature, F is Faraday’s constant, n is number of electrons involved, and i_0 is exchange-current density. From this equation the R_{ct} is related to i_0 , the larger the exchange current density, the smaller the interface charge transfer resistance. We conducted a half-cell experiment on the Zn electrode with a 3-electrode configuration to analyze the i_0 by Tafel curves. The i_0 depends critically on the nature of the electrode. The Zn(0002) exhibits the high exchange current density of 1.56 mA cm⁻², whereas the com-Zn presents the exchange current density of 0.64 mA cm⁻², indicating the fast deposition kinetic of the Zn(0002) (**Supplementary Fig. 26**). Consequently, the charge transfer resistance of Zn(0002) electrode is lower than that of the com-Zn electrode due to its higher exchange current density.

Changes:

Besides, the lower R_{ct} of Zn(0002) compared to that of com-Zn was attributed to its higher exchange current density (i_0) (Supplementary Fig. 26), indicating the fast deposition kinetic of the Zn(0002) electrode. (In the Manuscript, Page 12)

According to the Butler-Volmer equation, the expression of the charge-transfer resistance (R_{ct}) changes info:

$$R_{ct} = \frac{RT}{nFi_0} \quad (1)$$

Where R is gas constant, T is temperature, F is Faraday's constant, n is number of electrons involved, and i_0 is exchange-current density. From this equation the R_{ct} is related to i_0 , the larger the exchange current density, the smaller the interface charge transfer resistance. We conducted a half-cell experiment on the Zn electrode with a 3-electrode configuration to analyze the i_0 by Tafel curves. The i_0 depends critically on the nature of the electrode. The Zn(0002) exhibits the high exchange current density of 1.56 mA cm^{-2} , whereas the com-Zn presents the exchange current density of 0.64 mA cm^{-2} , indicating the fast deposition kinetic of the Zn(0002). (In the Supporting Information, Page R17)

Supplementary Fig. 26 | Measurement of exchange current density of Zn electrode. Linear polarization curves of com-Zn and Zn(0002) electrodes by a 3-electrode configuration.

6. The authors argued that structural homogeneity in (0002) plane results in a lower formation of

Zn₄SO₄(OH)₆·5H₂O by-products. Indeed, presence of byproducts or chemical state of Zn surface over repeated cycling is critical issue to analyze the origin of Zn anode stability. Hence, it is highly recommended to provide X-ray photoelectron spectroscopy (XPS) data of the cycled Zn anodes.

Response:

We would like to express our sincere gratitude for the valuable feedback provided by the reviewer. In order to further elaborate the high stability of Zn(0002) during cycling, we have included X-ray photoelectron spectroscopy (XPS) data of the cycled Zn electrodes. The adventitious carbon peak located at 284.8 eV was utilized for sample calibration. As presented in **Supplementary Fig. 22a**, the S 2*p* XPS spectra indicate distinct SO₄²⁻ peaks at 168.7 eV on the surface of Zn anodes after cycling, with significantly higher intensity observed on bare Zn. Meanwhile, the Zn 2*p*_{3/2} XPS spectra can be deconvoluted into two peaks at 1023.3 eV and 1021.8 eV, corresponding to Zn²⁺ and Zn⁰, respectively. Notably, the proportion of Zn²⁺ for com-Zn (20.3%) exceeds that in the Zn(0002) alloy (11.3%). Combined with XRD (**Supplementary Fig. 21**) results and EPMA-WDS images (**Supplementary Fig. 23a,b**) confirm the identification of the by-product resulting from parasitic reactions as zinc hydroxysulfate (Zn₄SO₄(OH)₆·5H₂O). These findings suggest the Zn(0002) metal electrode shows superior corrosion resistance than com-Zn electrode.

Changes:

This result was consistent with the XPS spectra and EPMA wavelength-dispersive X-ray spectroscopy (WDS) images of the cycled Zn electrode, as depicted in **Supplementary Fig. 22a,b and Supplementary Fig. 23a,b**. (In the Manuscript, Page 11)

The application of further XPS analysis unveils the surface chemical composition. The S 2*p* XPS spectra indicate distinct SO₄²⁻ peaks at 168.7 eV on the surface of Zn anodes after cycling, with significantly higher intensity observed on bare Zn. Meanwhile, the Zn 2*p*_{3/2} XPS spectra can be deconvoluted into two peaks at 1023.3 eV and 1021.8 eV, corresponding to Zn²⁺ and Zn⁰, respectively. Notably, the proportion of Zn²⁺ for com-Zn (20.3%) exceeds that in the Zn(0002) alloy (11.3%). These findings suggest the Zn(0002) metal electrode shows superior corrosion resistance than com-Zn electrode. (In the Supporting Information, Page R15)

Supplementary Fig. 22 | (a) Zn 2p, (b) S 2p XPS spectra of Zn electrodes after cycling.

Reviewer #2:

This manuscript reports the utilization of electrodeposited [0001]-uniaxial oriented Zn(0002) metal anodes in aqueous zinc-ion batteries. The authors claim that this Zn(0002) metal anode exhibits exceptional epitaxial growth, even under challenging conditions such as high depth of discharge and/or high areal capacity. While the authors have made efforts to enhance the electrochemical performance and have conducted various analyses to validate their claim, previous report has already explored the use of a single crystal Zn metal anode, and the evidence provided to support the author's claim is unreliable and often confusing. Therefore, considering the esteemed reputation of this journal, I do not support the publication of this manuscript in Nature Communications. Please refer to the comments below.

Response:

Thanks a lot for the careful reading and insightful suggestions to this manuscript, which are very helpful in improving the overall quality of our work. Firstly, we apologize for the lack of specificity in describing the preparation process of the single Zn(0002)-textured electrodes in the initial manuscript, resulting from our inadequate and deficient descriptions. In response to the reviewers' feedback, we have provided a more detailed and comprehensive elaboration on the essential requirements for creating a single (0002)-textured Zn metal electrode. This involves systematic modification of various variables within the electroplating preparation process, including the electrolyte components, current density and pH value.

Secondly, we notice that the reviewer expressed more concern regarding the preparation techniques and parameter control in the electrodeposition process, rather than emphasizing the importance of Zn metal electrodes possessing a single Zn(0002) texture for epitaxial sustainability. We recognize that this crucial aspect was not adequately highlighted in our initial manuscript. Consequently, we conducted additional experiments by preparing Zn metal electrodes with a relative texture coefficients (RTCs) of 93 for the Zn(0002) lattice plane as intermediate states (designated as IMS-Zn(0002)) to further elucidate the central focus of our manuscript. Our findings revealed that, with an increase in deposition time, the (0002) texture of the IMS-Zn(0002) metal electrode could only be sustained for a brief period, posing challenges in inducing continuous Zn growth along the [0001]_{Zn} direction.

Furthermore, it is crucial to note that our work differs significantly from the previously reported

single crystal Zn metal work (*Adv. Mater.* 2022, 34, 2202552) in terms of application perspective, regulation mechanism, and research direction. Additional details are presented in question 1.

By clarifying these issues that were not adequately clarified in the previous version, we sincerely hope that you will reconsider the publication of this work in *Nature Communications*.

1. The most concerning issue of this work is its lack of novelty. A previous report has already demonstrated that a single crystal (002) Zn metal anode in ZnSO₄ aqueous electrolyte exhibits remarkably high electrochemical performance compared to the use of other substrate materials, including polycrystalline Zn [*Adv. Mater.* 2022, 34, 2202552].

Response:

Thank you for the reviewer's suggestion; however, we believe that the work on single crystal (002) Zn work (*Adv. Mater.* 2022, 34, 2202552) mentioned by the reviewer differs significantly from our work. Firstly, compared to the exorbitant cost of single-crystal Zn, the electrodeposited Zn(0002) metal electrodes with a single [0001]-crystalline orientation are cost-effective and suitable for large-scale preparation. It's crucial to notice that one of the primary advantages of aqueous zinc-ion batteries (AZIBs) is their low cost. Directly purchasing single crystal Zn as anodes, as done in this single crystal (002) Zn work, would significantly diminish the cost-effectiveness of AZIBs. Therefore, these two approaches are incomparable in terms of practical application.

Secondly, the single crystal Zn work is focused on ensuring no defective shallow-angle grain boundaries in Zn metal electrode, which could eliminate a significant source of non-planar Zn nucleation. In contrast, our work aims to demonstrate the significance of a Zn metal electrode with 100% (0002) metal texture for achieving ultra-sustainable Zn-Zn homoepitaxial growth, eliminating Zn dendrite from the source. While single crystal Zn can meet the single Zn(0002)-texture requirements, it is relatively limited. Our research confirms that as long as each grain in the Zn metal electrode is along the [0001]_{Zn} orientation, it ensures that the subsequently deposited Zn presents flat Zn deposition. This means that a single crystal (002) Zn is an extreme case of a single (0002)-textured Zn.

Thirdly, the reviewer referred to the single crystal (002) Zn metal anode in ZnSO₄ electrolyte exhibits remarkably high electrochemical performance compared to the use of other substrate materials, including polycrystalline Zn. However, the polycrystalline Zn proposed in this work is completely

different from the Zn(0002) metal electrode prepared in our work. While our Zn(0002) metal electrode is also polycrystalline Zn, each grain's crystalline orientation is consistent, exposing the (0002) crystal plane (**Fig. 2e-i and Fig. R3b**), whereas the polycrystalline Zn constructed in the single crystal Zn work is disordered (**Fig. R3a**), making a direct comparison inappropriate.

Furthermore, the aforementioned single-crystal work merely demonstrates the excellent performance of a Zn metal anode with a single (0002) texture, without delving into the profound investigation of its epitaxial growth mechanism. In contrast, our work's major contribution lies in the use of more specialized and precise metal characterization techniques, such electron backscatter diffraction (EBSD), transmission electron microscopy (TEM), and even first applied high-angle angular dark-filed scanning transmission electron microscopy (HAADF-STEM), to fully substantiate the significance of constructing Zn metal anodes with 100% (0002) texture for achieving ultra-sustainable epitaxial growth of Zn (**Fig. 2e-i and Fig. 4e-h**). Our findings indicate that any presence of other textures in the Zn(0002) metal electrode, even in small amounts, will render the Zn-Zn epitaxial mechanism ineffective.

In summary, our work differs significantly from the single crystal Zn work mentioned by the reviewer. It not only presents a cost-effective strategy for constructing a single textured metallic electrode, suppressing dendrite formation and side reactions, but also offers new insights and theoretical guidance for reshaping electrodes in other metals. Additionally, our work will establish new benchmarks of texture regulation on Zn metal electrodes by focusing on achieving a single (0002) texture, rather than merely enhancing the (0002)-textured proportion in the future.

Fig. R3. XRD of (a) the electroplated Zn on various electrode substrates in the single crystal Zn work and (b) the com-Zn and Zn(0002) metal electrodes in our work.

2. The authors employed an $H_3BO_3/ZnSO_4$ -based solution to fabricate the Zn(0002) metal anode. However, upon examining the XRD results from the reference paper [Ref. 31], it becomes evident that despite Zn being electrodeposited in an electrolyte with the same components, additional crystallographic orientations coexist with (0002). In comparison to the previous paper, how does this manuscript account for the exclusive growth of Zn in the (0002) direction? Considering that the reference paper also utilized SO_4^{2-} anion in its electrolyte but displayed orientations other than (0002), it appears that factors such as stirring or high current density, which differ from the conditions in that paper, exert a more dominant influence on determining the preferred deposition orientation of Zn, surpassing the impact of the anion type.

Response:

Thanks for your careful reading and crucial question. In the reference paper [Ref.31], H_3BO_3 additive was employed in $ZnSO_4$ -based electrolyte as a side reaction inhibitor to regulate surface reaction kinetics of Zn metal anode. The single (0002)-textured Zn metal electrode was electrodeposited on Cu foil under a current density of 30 mA cm^{-2} in solutions containing 100 g L^{-1} $ZnSO_4 \cdot 6H_2O$ and 20 g L^{-1} H_3BO_3 , with a pH value of 2 and vigorous agitation (700 r min^{-1}). It is important to note that the vigorous stirring during the electrodeposition preparation process serves to eliminate H_2 generated during the Zn plating process, preventing gas accumulation on the substrate and enhancing deposit quality (*Surface & Coating Technology, 2006, 201, 371-383; Surface & Coating Technology, 2002, 157, 282-289*). Moreover, intense stirring guarantees that Zn deposition remains unaffected by the substrate, as it minimizes the contact duration between the zinc ions and the substrate. Boric acid (H_3BO_3), as mentioned by the reviewer, is a commonly used buffer agent in acidic deposition solutions, primarily serving to stabilize the pH value (*Journal of Pharmaceutical Science, 2020, 2375-2386; Journal of the Electrochemical Society, 2020, 167, 11250*). As described in the specialized electroplating manual by Kanani N. (*Electroplating: basic principles, processes and practice. Elsevier. 2004*), the presence of H_3BO_3 ensures the relative stability of the pH value in the plating solution. This is attributed to H_3BO_3 being a triprotic weak acid, capable of dissociating or absorbing hydrogen ions in response to fluctuations in the plating solution's pH ($H_3BO_3 \leftrightarrow H_2BO_3^- + H^+$; $H^+ + OH^- \leftrightarrow H_2O$).

To attain a more thorough comprehension of the impact of various electroplating parameters,

including H_3BO_3 additive, pH value and current density, on the metal texture of electrodeposited Zn electrodes, a series of experiments were conducted (**Supplementary Table 1**), as detailed below:

i) Under 30 mA cm^{-2} current density and vigorous stirring, the electrodeposited Zn metal electrodes in 100 g L^{-1} ZnSO_4 solution led to the formation of $\text{Zn}_4\text{SO}_4(\text{OH})_6 \cdot 5\text{H}_2\text{O}$ (ZHS) by-products on its surface due to severe side reactions, resulting in a disordered Zn texture (**Supplementary Fig. 1a, Supplementary Fig. 2a and Supplementary Fig. 3a**).

ii) Under 30 mA cm^{-2} current density and vigorous stirring, the electrodeposited Zn metal electrodes in 100 g L^{-1} ZnSO_4 solution with $\text{pH} = 2$ can maintain a single crystalline orientation in the initial 10 min. However, with an increase in deposition time, the deposition solution experienced a decrease in pH value, triggering enhanced side reactions and a reduction in deposition efficiency. Consequently, this phenomenon resulted in the formation of a disordered texture and an uneven surface on the prepared Zn electrodes (**Supplementary Fig. 1b, Supplementary Fig. 2b and Supplementary Fig. 3b**).

iii) Under 30 mA cm^{-2} current density and vigorous stirring, the electrodeposited Zn metal electrodes in 100 g L^{-1} ZnSO_4 and 20 g L^{-1} H_3BO_3 solution with $\text{pH} = 2$ presented a single Zn(0002) texture without any by-products (**Supplementary Fig. 1c, Supplementary Fig. 2c and Supplementary Fig. 3c**). This result can be attributed to the incorporation of H_3BO_3 and vigorous stirring. These factors collectively create an ideal environment, enabling maximum exposure of the Zn(0002) crystal plane with the lowest surface energy, while remaining unaffected by the substrate.

iv) Regulating the current density within the parameters of Experiment iii revealed intriguing findings. At lower currents (10 and 20 mA cm^{-2}), it became feasible to fabricate zinc metal electrodes with a single (0002) texture (**Supplementary Fig. 3d**). However, this achievement was accompanied by reduced deposition efficiency and surface unevenness (**Supplementary Fig. 1d and Supplementary Fig. 2d**). Conversely, higher current density (50 mA cm^{-2}) exacerbated side reactions and uncontrolled Zn growth, leading to a disordered Zn texture (**Supplementary Fig. 2d and Supplementary Fig. 3d**). Consequently, a current density of 30 mA cm^{-2} appears to be the optimal electroplating condition.

In summary, stirring, H_3BO_3 additives, pH values, and appropriate current density are essential factors in creating an optimal electroplating environment for fabricating a single (0002)-textured Zn metal electrode. Therefore, in this ideal environment, within the ZnSO_4 system, as opposed to the

ZnCl₂ and Zn(CH₃COO)₂ systems, the surface energy of the Zn(0002) crystal plane undergoes the most significant reduction (**Fig. 1e,f**), making it easier to exposure.

Changes:

This was achieved through the incorporation of H₃BO₃ additive and vigorous stirring, creating an ideal environment to eliminate harmful side reactions³¹⁻³⁵. As a result, the crystallographic orientation and texture of the electrodeposited Zn metals were exclusively dependent on the anions present in the deposited liquids. Detailed experimental parameter regulation is provided in **Supplementary Table 1** and **Supplementary Fig. 1-3**. The Zn metals deposited from ZnCl₂ and Zn(CH₃COO)₂ systems displayed irregular crystalline orientation, whereas a single (0002)-textured Zn metal was obtained from the ZnSO₄ system, as shown in the scanning electron microscopy (SEM) images (**Fig. 1b-d**). (In the Manuscript, Page 4-5)

Supplementary Table 1. The electrodeposition parameters of Zn metal electrodes

Experiment number	i	ii	iii	iv
Electrodeposition parameters	100 g L ⁻¹ ZnSO ₄ ·6H ₂ O, J = 30 mA cm ⁻²	100 g L ⁻¹ ZnSO ₄ ·6H ₂ O, pH = 2 , J = 30 mA cm ⁻²	100 g L ⁻¹ ZnSO ₄ ·6H ₂ O, pH = 2, 20 g L⁻¹ H₃BO₃ , J = 30 mA cm ⁻²	100 g L ⁻¹ ZnSO ₄ ·6H ₂ O, pH = 2, 20 g L ⁻¹ H ₃ BO ₃ , J = 10~50 mA cm⁻²

Supplementary Fig. 1. Optical images of prepared Zn metal electrodes under the corresponding electrodeposition parameters of Supplementary Table 1. Experiment (a) i, (b) ii, (c) iii and (d) iv.

Supplementary Fig. 2. SEM images of prepared Zn metal electrodes under the corresponding

electrodeposition parameters of Supplementary Table 1. Experiment (a) i, (b) ii, (c) iii and (d) iv.

Supplementary Fig. 3. XRD patterns of prepared Zn metal electrodes under the corresponding electrodeposition parameters of Supplementary Table 1. Experiment (a) i, (b) ii, (c) iii and (d) iv.

To attain a more thorough comprehension of the impact of various electroplating parameters, including H_3BO_3 additive, pH value, and current density, on the metal texture of electrodeposited Zn electrodes, a series of experiments were conducted (**Supplementary Table 1**), as detailed below:

i) Under 30 mA cm^{-2} current density and vigorous stirring, the electrodeposited Zn metal electrodes in 100 g L^{-1} ZnSO_4 solution led to the formation of $\text{Zn}_4\text{SO}_4(\text{OH})_6 \cdot 5\text{H}_2\text{O}$ (ZHS) by-products on its surface due to severe side reactions, resulting in a disordered Zn texture (**Supplementary Fig. 1a**, **Supplementary Fig. 2a** and **Supplementary Fig. 3a**).

ii) Under 30 mA cm^{-2} current density and vigorous stirring, the electrodeposited Zn metal electrodes in 100 g L^{-1} ZnSO_4 solution with $\text{pH} = 2$ can maintain a single crystalline orientation in the initial 10 min. However, with an increase in deposition time, the deposition solution experienced a

decrease in pH value, triggering enhanced side reactions and a reduction in deposition efficiency. Consequently, this phenomenon resulted in the formation of a disordered texture and an uneven surface on the prepared Zn electrodes (**Supplementary Fig. 1b, Supplementary Fig. 2b and Supplementary Fig. 3b**).

iii) Under 30 mA cm^{-2} current density and vigorous stirring, the electrodeposited Zn metal electrodes in $100 \text{ g L}^{-1} \text{ ZnSO}_4$ and $20 \text{ g L}^{-1} \text{ H}_3\text{BO}_3$ solution with $\text{pH} = 2$ presented a single Zn(0002) texture without any by-products (**Supplementary Fig. 1c, Supplementary Fig. 2c and Supplementary Fig. 3c**). This result can be attributed to the incorporation of H_3BO_3 and vigorous stirring. These factors collectively create an ideal environment, enabling maximum exposure of the Zn(0002) crystal plane with the lowest surface energy, while remaining unaffected by the substrate.

iv) Regulating the current density within the parameters of Experiment iii revealed intriguing findings. At lower currents (10 and 20 mA cm^{-2}), it became feasible to fabricate zinc metal electrodes with a single (0002) texture (**Supplementary Fig. 3d**). However, this achievement was accompanied by reduced deposition efficiency and surface unevenness (**Supplementary Fig. 1d and Supplementary Fig. 2d**). Conversely, higher current density (50 mA cm^{-2}) exacerbated side reactions and uncontrolled Zn growth, leading to a disordered Zn texture (**Supplementary Fig. 2d and Supplementary Fig. 3d**). Consequently, a current density of 30 mA cm^{-2} appears to be the optimal electroplating condition.

In summary, stirring, H_3BO_3 additives, pH values, and appropriate current density are fundamental factors in establishing an optimal electroplating environment to fabricate a single (0002)-textured Zn metal electrode. (In the Supporting Information, Page R2-R5)

3. The authors used Cu as the substrate to prepare the Zn(0002) metal anode. Despite a significant lattice mismatch between the Cu substrate and the subsequently electrodeposited Zn, the (0002) directing of Zn was still achievable. This suggests that the composition of the electrolyte bath or the operating conditions of the electrodeposition process, independent of lattice mismatch, act as dominant

factors in regulating the texture through different mechanisms. Therefore, there appears to be a contradiction between the claim of lattice mismatch in this paper and the observed phenomenon.

Response:

Thank you very much for raising this question. It was our oversight not to list the choice of deposition substrates, which led to this misunderstanding. During the preparation process of Zn metal electrodes, we were able to effectively suppress side reactions such as corrosion and hydrogen evolution by using boric acid, vigorous stirring, and a fixed current density. Building upon this foundation, we have created an ideal environment for electrodepositing Zn, ensuring maximal surface exposure with the lowest surface energy, unaffected by the substrate. During the initial substrate screening process, we successfully prepared Zn metals with a single (0002) texture on stainless steel (amorphous), Cu foil, and Al foil substrates, respectively. In contrast to Zn deposition within the internal battery, vigorous stirring guarantees that Zn deposition remains unaffected by the substrate, as it minimizes the contact duration between the zinc ions and the substrate. Therefore, regardless of whether the substrate is crystalline or amorphous, the electrodeposited Zn exhibited a single (0002) texture, unaffected by the underlying substrate texture. Considering the adhesion strength with the Zn electroplating layer, we ultimately selected Cu foil as the electrodeposition substrate (**Fig. R4a,b**). The circumstances encountered throughout the entire preparation process and the internal deposition within the battery are completely different and should not be compared. Inside the battery, the prepared single (0002)-textured Zn metal electrode exhibits superior thermodynamic stability and corrosion resistance to suppress side reactions. The subsequently deposited Zn grows along the crystalline orientation of the substrate.

Fig. R4. Screening of Zn-deposited substrates. (a) Optical micrograph of the Zn electrodeposits on

stainless steel, Al foil, and Cu foil substrates. (b) Complementary view of the opposite side of the electrodeposits shown in (a). (c) XRD patterns of the Zn electrodeposits on the three substrates.

4. The authors claim anion-induced texture regulation of Zn crystals; however, the comparative results they provided based on different anions only include calculations of surface energy and XRD data of the electrode. However, there is a possibility that substances like boric acid present in the electrolyte bath during the preparation of the Zn(0002) metal anode could have adsorbed onto the surface or caused surface modifications such as SEI formation. Therefore, to truly validate the effects resulting from differences in the deposition crystallographic orientation due to anions, it would be necessary to present electrochemical data using electrodes deposited under the same conditions, except for using different anions as a control group, rather than relying solely on commercial Zn. This would provide substantial evidence to substantiate the influence of anions on the deposition orientation, beyond mere speculation.

Response:

Thank you for the constructive suggestion. In light of the reviewers' inquiries, we notice that the reviewer expressed more concern regarding the preparation techniques and parameter control in the electrodeposition process, rather than the primary focus on the emphasizing the importance of Zn metal electrodes possessing a single Zn(0002) texture for epitaxial sustainability. We recognize that this crucial aspect was not adequately highlighted in our initial manuscript. Consequently, we conducted additional experiments by preparing Zn metal electrodes with a relative texture coefficients (RTCs) of 93 for the Zn(0002) lattice plane as intermediate states (designated as IMS-Zn(0002)) to further elucidate the central focus of our manuscript. The IMS-Zn(0002) metal electrode was synthesized by incorporating 5 g/L $\text{Zn}(\text{CH}_3\text{COO})_2 \cdot 2\text{H}_2\text{O}$ into the deposited solution employed for the preparation of the Zn(0002) metal electrode.

The microstructure of IMS-Zn(0002) metals was systematically investigated through XRD, SEM and electron backscatter diffraction (EBSD) observations. The IMS-Zn(0002) metal electrode, featuring an average grain size value of $8.4 \pm 0.3 \mu\text{m}$, showed a hexagonal morphology similar to that of the Zn(0002) metal electrode (**Supplementary Fig. 9a** and **Supplementary Fig. 10a,b**). The XRD pattern (**Supplementary Fig. 9b**) of the IMS-Zn(0002) electrode displayed a highly pronounced (002) peak and a weak (101) peak. The relative texture coefficient (RTC) of each lattice plane was calculated

using the following formula,

$$RTC_{(hkl)} = \frac{I_{(hkl)}/\bar{I}_{(hkl)}}{\sum[I_{(hkl)}/\bar{I}_{(hkl)}]} \times 100 \quad (2)$$

Where, $I_{(hkl)}$ represents the intensity obtained from textured Zn sample, and $\bar{I}_{(hkl)}$ is the intensity of the standard oriented Zn sample (from JCPDS data). The corresponding $RTC_{(002)}$ of IMS-Zn(0002) was 93 (**Supplementary Fig. 11**). This result indicates that the IMS-Zn(0002) metal electrodes had only a minor (10 $\bar{1}$ 1) texture predominantly featuring the (0002) texture, which was further confirmed by the EBSD characterization (**Supplementary Fig. 12a-d**).

To further observe the morphology evolution of Zn metal electrodes during plating/stripping, a series of in situ and ex situ tests were carried out. On the stripping side, both com-Zn and IMS-Zn(0002) electrodes exhibited numerous randomly-sized pores, which became more pronounced with discharge time (**Supplementary Fig. 13a,b**). Corrosion by-products were even formed on the com-Zn surface. In contrast, the Zn(0002) surface displayed greater corrosion resistance and a more ordered stripping process (**Supplementary Fig. 13c**). On the plating side, scattered and loose Zn deposits were observed on the com-Zn surface, which rapidly grew along the separator direction (**Fig. 3a**). The IMS-Zn(0002) electrode maintained a relatively flat deposition surface during the 10 min plating. However, with an increase in deposition time, disordered and uneven Zn deposits formed on the IMS-Zn(0002) surface (**Fig. 3c**). In contrast, hexagonal-Zn deposition spread parallelly without Zn dendrite on the Zn(0002) surface during plating (**Fig. 3e**). The IMS-Zn(0002) electrode maintained a relatively flat deposition surface during the 10 min plating. However, with an increase in deposition time, disordered and uneven Zn deposits formed on the IMS-Zn(0002) surface (**Fig. 3c**). In contrast, hexagonal-Zn deposition spread parallelly without Zn dendrite on the Zn(0002) surface during plating (**Fig. 3e**). Even after 1h plating, flat and densely packed Zn deposition was observed on the Zn(0002) surface using confocal laser microscope (LSM, **Supplementary Fig. 15a-c**), with a lower surface roughness (0.312 μm) than on com-Zn (2.8 μm) and IMS-Zn(0002) (1.9 μm) surfaces. Ex situ XRD showed that the deposited Zn on com-Zn and IMS-Zn(0002) surfaces lacked a fixed orientation, while a single (002) peak was consistently observed using the Zn(0002) electrode from the beginning to the end, illustrating ultra-sustainable epitaxial growth between the overgrowth and the substrate crystals. Furthermore, the (002) peak intensity of the deposited Zn using the Zn(0002) electrode was stronger, further demonstrating their perfect lattice match. These findings suggest that while employing a Zn metal electrode with a

predominant (0002) texture promotes uniform Zn deposition at the initial stage, the subsequently deposited Zn shows irregular and random growth once the deposition thickness surpasses the critical thickness (t_c) of epitaxial growth.

As a result, even a minor lattice mismatch between the deposits and substrate can induce the failure of the epitaxial mechanism. Therefore, it is imperative to ensure the Zn(0002) metal substrate with a single crystalline orientation to eliminate the lattice mismatch at epitaxial interface and provide uninterrupted driving forces for Zn epitaxial growth.

Changes:

Besides, to investigate the significance of a Zn metal electrode possessing a single (0002) texture for sustaining epitaxial growth, we constructed Zn metal electrodes with a predominant (0002) texture and a minor presence of other Zn textures as an intermediate state (IMS-Zn(0002)) between com-Zn and single (0002)-textured Zn electrodes. The IMS-Zn(0002) metal electrode, featuring an average grain size value of $8.4 \pm 0.3 \mu\text{m}$, showed a hexagonal morphology similar to that of the Zn(0002) metal electrode (**Supplementary Fig. 9a** and **Supplementary Fig. 10a,b**). The XRD pattern (**Supplementary Fig. 9b**) of the IMS-Zn(0002) electrode displayed a highly pronounced (002) peak and a weak (101) peak. The relative texture coefficient (RTC) of each lattice plane was calculated using the following formula⁴⁷,

$$RTC_{(hkl)} = \frac{I_{(hkl)}/\bar{I}_{(hkl)}}{\sum [I_{(hkl)}/\bar{I}_{(hkl)}]} \times 100 \quad (1)$$

Where, $I_{(hkl)}$ represents the intensity obtained from textured Zn sample, and $\bar{I}_{(hkl)}$ is the intensity of the standard oriented Zn sample (from JCPDS data). The corresponding $RTC_{(002)}$ of com-Zn, IMS-Zn(0002) and Zn(0002) electrodes was 41, 93 and 100, respectively (**Supplementary Fig. 11**). This result indicates that the IMS-Zn(0002) metal electrodes had only a minor ($10\bar{1}1$) texture predominantly featuring the (0002) texture, which was further confirmed by the EBSD characterization (**Supplementary Fig. 12a-d**). (In the Manuscript, Page 7)

The morphology evolution of com-Zn, IMS-Zn(0002) and Zn(0002) metal electrodes during plating and stripping was characterized by a series of in situ and ex situ tests. On the stripping side, both com-Zn and IMS-Zn(0002) electrodes exhibited numerous randomly-sized pores, which became more pronounced with discharge time (**Supplementary Fig. 13a,b**). Corrosion by-products were even formed on the com-Zn surface. In contrast, the Zn(0002) surface displayed greater corrosion resistance

and a more ordered stripping process (**Supplementary Fig. 13c**). On the plating side, scattered and loose Zn deposits were observed on the com-Zn surface, which rapidly grew along the separator direction (**Fig. 3a**). The IMS-Zn(0002) electrode maintained a relatively flat deposition surface during the 10 min plating. However, with an increase in deposition time, disordered and uneven Zn deposits formed on the IMS-Zn(0002) surface (**Fig. 3c**). In contrast, hexagonal-Zn deposition spread parallelly without Zn dendrite on the Zn(0002) surface during plating (**Fig. 3e**). (In the Manuscript, Page 7-8)

Even after 1h plating, flat and densely packed Zn deposition was observed on the Zn(0002) surface using confocal laser microscope (LSM, **Supplementary Fig. 15a-c**), with a lower surface roughness ($0.312\ \mu\text{m}$) than on com-Zn ($2.8\ \mu\text{m}$) and IMS-Zn(0002) ($1.9\ \mu\text{m}$) surfaces. Ex situ XRD was conducted to detect the structural evolution of Zn metal electrodes during the charging process, as shown in **Fig. 3b,d,f** and **Supplementary Fig. 16a-c**. The peaks located at approximately 36° in both electrodes were assigned to the (0002) plane of Zn crystal⁴⁸. The XRD patterns showed that the deposited Zn on com-Zn and IMS-Zn(0002) surfaces lacked a fixed orientation, while a single (002) peak was consistently observed using the Zn(0002) electrode from the beginning to the end, illustrating ultra-sustainable epitaxial growth between the overgrowth and the substrate crystals. Furthermore, the (002) peak intensity of the deposited Zn using the Zn(0002) electrode was stronger, further demonstrating their perfect lattice match. These findings suggest that while employing a Zn metal electrode with a predominant (0002) texture promotes uniform Zn deposition at the initial stage, the subsequently deposited Zn shows irregular and random growth once the deposition thickness surpasses the t_c of epitaxial growth. (In the Manuscript, Page 8)

The Zn metal with a relative texture coefficients (RTCs) of 93 for the Zn(0002) lattice plane was synthesized by adding 5 g/L $\text{Zn}(\text{CH}_3\text{COO})_2 \cdot 2\text{H}_2\text{O}$ in the above solution. (In the Manuscript, Page 13)

Fig. 3 | Morphologies and orientation of Zn plating and stripping. a,c,e SEM images of (a) com-Zn, (c) IMS-Zn(0002) and (e) Zn(0002) electrodes in Zn||Zn cells after plating 0 to 30 min (scale bar, 20 μm). b,d,f XRD patterns of (b) com-Zn, (d) IMS-Zn(0002) and (f) Zn(0002) electrodes after plating from 0 to 30 min. Current density, $J = 4 \text{ mA cm}^{-2}$.

Supplementary Fig. 9 | (a) SEM image and (b) XRD pattern of the electrodeposited IMS-Zn(0002) metal electrode.

Supplementary Fig. 10 | (a) SEM image of the polished IMS-Zn(0002) metal electrode. (b) Grain size distributions from statistical SEM measurement for the IMS-Zn(0002) metal electrode.

Supplementary Fig. 11 | The calculated $RTC_{(002)}$ value of com-Zn, IMS-Zn(0002) and Zn(0002) metal electrodes.

Supplementary Fig. 12 | Orientation maps of IMS-Zn(0002) metals along (a) normal direction (ND), (b) rolling direction (RD) and (c) transverse direction (TD). (d) The corresponding (0002) pole figure.

Supplementary Fig. 13 | SEM images of (a) com-Zn and (b) Zn(0002) electrodes after stripping from 5 to 30 min (scale bar, 20 μm). Current density, $J = 4 \text{ mA cm}^{-2}$.

Supplementary Fig. 15 | LSM images of (a) com-Zn, (b) IMS-Zn(0002) and (c) as-deposited Zn(0002) electrodes after plating 1 h.

Supplementary Fig. 16 | XRD patterns of Zn electrodes after plating 0, 5, 10, 20, 30 min, respectively. (a) com-Zn, (b) IMS-Zn(0002) and (c) Zn(0002).

REVIEWER COMMENTS

Reviewer #1 (Remarks to the Author):

This manuscript reports an electrochemical fabrication of the homoepitaxial Zn(0002) surface and analyses of its crystallographic effect on reversible Zn anode cyclability. The authors demonstrated successful electrodeposition of (0002) plane-rich Zn film on Cu substrate in H₃BO₃ and sulfate anion under vigorous mass transfer conditions. In response to the reviewer's points, the authors effectively supported their argument with additional experiments, including parameter effects in (0002) Zn film electrodeposition, comparison of hydrogen evolution reaction rate, and discussion of charge transfer resistance. With additional supporting experiments, the overall argument becomes more clear and reasonable from the reviewer's side. Therefore, the reviewer recommends the revised manuscript to be suitable for publication in *Nature Communications*.

Reviewer #2 (Remarks to the Author):

I appreciate the authors' efforts in addressing the concerns raised in the initial review and presenting a well-structured rebuttal. However, after careful consideration of the revised manuscript, I still find significant issues that need to be addressed before the publication of this work in *Nature Communications*.

1. The authors argue that the morphology differences in zinc metal deposition based on the type of anion are due to variations in surface energy resulting from anion adsorption. However, the authors claim that vigorous stirring minimizes the contact duration between zinc ions and the substrate, leading to a uniform (0002) texture regardless of substrate type. In such situations where vigorous stirring has a pronounced effect, attributing epitaxial deposition solely to anion adsorption based on calculated surface energy differences becomes questionable. It is crucial to explore whether variations in ion solvation structures or adjustments made to reach pH 2—where the use of different salts may result in different initial pH values—could contribute to the observed effects. Detailed explanations and additional evidence supporting the results are essential to substantiate the authors' central argument.
2. The main assertion of this manuscript is that epitaxial deposition is sustained only when exposing 100% (0002) facets (Zn(0002) electrode). However, the reasons why the effects cannot be maintained when other facet orientations are present have not been sufficiently explained or supported with evidence. While the introduction briefly mentions the concept of critical thickness, a thorough explanation is missing, and the specific correlation of this concept with minor lattice mismatch and epitaxial deposition failure is not established. In situations where this mechanism has not been fully understood, the inability to achieve epitaxial deposition on other facets of Zn, where lattice mismatch can be minimal, raises questions that need clarification.
3. The authors report that even with ZnSO₄, approximately 30% of orientations other than (0002) exist based on calculations. This discrepancy, considering the significant theoretical value, necessitates an explanation for how 100% (0002) texture is attainable in actual experimental conditions.
4. Why do the Zn(CH₃COO)₂ and ZnCl₂-based solutions, excluding the case that used ZnSO₄, exhibit overall XRD peak shifts in supplementary Fig.4?
5. In Supplementary Figure 19, even considering that side reactions may be relatively severe at low current densities, the performance of the control group seems to be significantly lower compared to the reported AZIBs. Please provide a detailed explanation on this discrepancy.
6. The manuscript predominantly focuses on high-rate performances. While high-rate performance is indeed an advantage of AZIBs, the issue of severe side-reactions at low rates is significant in this system. Given the authors' claim regarding the substantial reduction of corrosion and side reactions through the exposure of (0002) facets of Zn, it is essential to include electrochemical performance data at low current densities for both the Zn symmetric cell and full cell configuration. This information would provide a more comprehensive understanding of the electrode's behavior across a range of operating conditions.

7. Numerous instances of missing explanations, typos, and errors are present throughout the manuscript. For example, critical concepts, such as f_s in the equation of critical thickness ($t_c = b(f_s/f)^2$), require proper clarification. Additionally, in Supplementary Fig.13, there are incorrect descriptions for (b) and an omission for (c). Furthermore, in Supplementary Fig.29, contrary to the authors' assertion, the SEM image of (a) com-Zn appears to be more intact and uniform than (b) Zn(0002) metal anode. These instances, along with others, need careful correction and clarification for the manuscript to meet the required standard.

Response to Reviewers' Comments

We express sincere appreciation for the constructive feedback and insightful suggestions provided by the reviewers on our manuscript. We carefully considered these comments, conducted additional experiments, and revised the manuscript accordingly. Below is a detailed description of the changes, accompanied by our point-by-point response to the reviewers' comments.

Reviewer #1

This manuscript reports an electrochemical fabrication of the homoepitaxial Zn(0002) surface and analyses of its crystallographic effect on reversible Zn anode cyclability. The authors demonstrated successful electrodeposition of (0002) plane-rich Zn film on Cu substrate in H₃BO₃ and sulfate anion under vigorous mass transfer conditions. In response to the reviewer's points, the authors effectively supported their argument with additional experiments, including parameter effects in (0002) Zn film electrodeposition, comparison of hydrogen evolution reaction rate, and discussion of charge transfer resistance. With additional supporting experiments, the overall argument becomes more clear and reasonable from the reviewer's side. Therefore, the reviewer recommends the revised manuscript to be suitable for publication in *Nature Communications*.

Response:

We express sincere gratitude for the invaluable comments provided by the reviewer and their generous recommendation of our research.

Reviewer #2

I appreciate the authors' efforts in addressing the concerns raised in the initial review and presenting a well-structured rebuttal. However, after careful consideration of the revised manuscript, I still find significant issues that need to be addressed before the publication of this work in *Nature Communications*.

Response:

We sincerely appreciate the reviewer's acknowledgement of our revised manuscript and the

opportunity for further refinement and enhancement. The constructive feedback from the reviewer has substantially improved the overall quality of our paper. Furthermore, we highly value the reviewer's supplementary comments, and in response, we have provided further clarifications and made necessary revisions.

1. The authors argue that the morphology differences in zinc metal deposition based on the type of anion are due to variations in surface energy resulting from anion adsorption. However, the authors claim that vigorous stirring minimizes the contact duration between zinc ions and the substrate, leading to a uniform (0002) texture regardless of substrate type. In such situations where vigorous stirring has a pronounced effect, attributing epitaxial deposition solely to anion adsorption based on calculated surface energy differences becomes questionable. It is crucial to explore whether variations in ion solvation structures or adjustments made to reach pH 2—where the use of different salts may result in different initial pH values—could contribute to the observed effects. Detailed explanations and additional evidence supporting the results are essential to substantiate the authors' central argument.

Response:

Thanks for the reviewer's constructive comment. The acid electrolyte effectively guarantee the formation of Zn(0002) texture by suppressing side reactions. As shown in Fig. R1a-c, under 30 mA cm⁻² current density and vigorous stirring, the electrodeposited Zn metal electrodes in 100 g L⁻¹ ZnSO₄ solution (pH = 3.5) led to the formation of disordered Zn textures due to detrimental side reactions. While keeping other variables constant, lowering the pH value of the deposition solution to 2 facilitates maintaining a single (0002) texture in the Zn electrode during the initial 10 min (Fig. R2a-c). With an increase in deposition time, a decrease in the deposition solution's pH triggers side reactions, resulting in a disordered crystalline orientation of subsequently deposited Zn (Fig. R2c). However, the introduction of H₃BO₃ as a pH buffering agent ($H_3BO_3 \leftrightarrow H_2BO_3^- + H^+$; $H^+ + OH^- \leftrightarrow H_2O$) maintains a constant pH value of 2 throughout the entire electrodeposition process, ensuring a single [0001]-crystalline orientation in the deposited Zn (Fig. R3a-c). According to the professional electroplating manual by Kanani N. (*Electroplating: basic principles, processes and practice. Elsevier. 2004*), these results are attributable to the potential inadequate supply of Zn²⁺ ions at the substrate during the electrodeposition process, especially with increasing deposition time. This

shortage causes the reduction of H^+ ions by electrodes, leading to the release of H_2 . Subsequently, the pH in the cathodic region experiences a swift rise, promoting the precipitation of $Zn_4SO_4(OH)_6 \cdot 5H_2O$ (ZHS) by-products and yielding a coarse and nonplanar deposited Zn. Consequently, a constant pH value of 2 serves exclusively to inhibit side reactions on the deposition side and has no influence on the crystalline orientation of the deposited Zn.

Fig. R1. (a) Optical images, (b) SEM images and (c) XRD patterns of prepared Zn metal electrodes from deposited solution with an initial pH value of 3.5.

Fig. R2. (a) Optical images, (b) SEM images and (c) XRD patterns of prepared Zn metal electrodes from deposited solution with an initial pH value of 2.

Fig. R3. (a) Optical images, (b) SEM images and (c) XRD patterns of prepared Zn metal electrodes from deposited solution with a constant pH value of 2.

Additionally, we have not considered the impact of anion types on the solvation structure of Zn^{2+} ions. Typically, Zn^{2+} cations are octahedrally coordinated with six water molecules forming $[Zn(H_2O)_6]^{2+}$ with high charge density. In the event that anions interfere with the solvation structure of Zn^{2+} ions, the corresponding chemical formula and their calculated enthalpy changes (ΔH) are presented in Table R1.

Table R1. Calculated enthalpy changes for the desolvation reactions.

Reactions	ΔH (kJ/mol)
$[Zn(H_2O)_6]^{2+} + 2Cl^- \rightarrow ZnCl_2 + 6H_2O$	66.4
$[Zn(H_2O)_6]^{2+} + SO_4^{2-} \rightarrow ZnSO_4 + 6H_2O$	242.4
$[Zn(H_2O)_6]^{2+} + 2CH_3COO^- \rightarrow Zn(CH_3COO)_2 + 6H_2O$	21.4

Upon comparing the enthalpy changes, all three reactions involving $[Zn(H_2O)_6]^{2+}$ with Cl^- , SO_4^{2-} , and CH_3COO^- exhibit positive values, indicating their endothermic nature and unfavorable status under standard conditions. Consequently, the presence of anions (Cl^- , SO_4^{2-} , and CH_3COO^-) does not impact the solvation structure of Zn^{2+} ions.

In summary, the combination of experimental results and enthalpy calculations indicates that, under the condition of excluding side reactions, SO_4^{2-} anions demonstrate a superior ability to significantly reduce the surface energy of the (0002) crystal plane when compared to the other two anions (Cl^- and CH_3COO^-). Consequently, this results in the achievement of a Zn metal anode with a single (0002) texture. Furthermore, based on Gibbs-Wulff crystal growth theory ($\sigma_i/r_i = \text{constant}$, where σ_i represents the specific surface free energy of crystal face i , and r_i denotes the central distance from the equilibrium-formed crystal), it can be succinctly stated that crystals with lower

surface energy exhibit faster growth rates and become the primary exposed facets (*Advanced Materials*, 2016, 28, 1679-1702; *Mineral*, 1901, 34, 449). This theory further elucidates that, in the preparation process, the lowest surface energy of Zn(0002) plane in the ZnSO₄ system ultimately results in the single (0002)-textured Zn metal anodes.

Changes: Based on the Gibbs-Wulff theory of crystal growth, crystals with lower surface energy exhibit faster growth rates, consequently emerging as the primary exposed crystal planes³⁶. (In the Manuscript, Page 5)

[36] Li, R. et al. Gibbs–Curie–Wulff theorem in organic materials: a case study on the relationship between surface energy and crystal growth. *Adv. Mater.* **28**, 1697-1702 (2016). (In the Manuscript, Page 21)

2. The main assertion of this manuscript is that epitaxial deposition is sustained only when exposing 100% (0002) facets (Zn(0002) electrode). However, the reasons why the effects cannot be maintained when other facet orientations are present have not been sufficiently explained or supported with evidence. While the introduction briefly mentions the concept of critical thickness, a thorough explanation is missing, and the specific correlation of this concept with minor lattice mismatch and epitaxial deposition failure is not established. In situations where this mechanism has not been fully understood, the inability to achieve epitaxial deposition on other facets of Zn, where lattice mismatch can be minimal, raises questions that need clarification.

Response:

We sincerely appreciate the reviewer's insightful question. We apologize for not adequately explaining the ultra-sustainable homoepitaxial mechanism of the manuscript. As illustrated in Supplementary Fig. 17a, if there is a tolerable lattice mismatch between the substrate crystal and the overgrowth crystal, and the substrate crystal possesses a single crystalline orientation, the subsequently deposited crystals exhibit the anticipated crystalline orientation. This phenomenon has been previously documented in the context of heteroepitaxial preparation of Zn(0002) metal anodes, as evidenced in scientific literature (*Science*, 2019, 366, 654-648). However, in our study, we emphasize that the prepared Zn(0002) metal anodes only possess a singular crystalline orientation to achieve ultra-sustainable homoepitaxial growth. The presence of even a small amount of other Zn textures in the Zn(0002) metal anode will result in the formation of dislocations (as shown in the orange region in Supplementary Fig. 17b), leading to lattice distortion. Once the thickness of the deposition layer surpasses the critical value for epitaxial growth, the subsequently deposited crystals

display a disordered crystalline orientation, indicating the failure of the epitaxial mechanism. To underscore the significance of preparing Zn metal anodes with a 100% (0002) texture for achieving ultra-sustainable epitaxial growth of Zn, we examined residual stresses before and after deposition on Zn(0002) ($RCT_{(0002)} = 100\%$), IMS-Zn(0002) ($RCT_{(0002)} = 93\%$), and com-Zn metal electrodes. There is a significant increase in stress for both IMS-Zn(0002) (from 16.5 MPa to 60.8 MPa) and com-Zn (from 22.8 MPa to 110.7 MPa) metal electrodes before and after deposition. In contrast, the stress of Zn(0002) metal anodes change slightly from 14.8 MPa to 15.7 MPa, indicating negligible lattice strain and mismatch dislocations upon subsequent Zn deposition on the surface of Zn(0002) metal electrode. This observation further confirms the sustained epitaxial growth of Zn(0002) metal electrode. Therefore, it is imperative to ensure the Zn(0002) metal substrate with a single crystalline orientation to eliminate the lattice mismatch at epitaxial interface and provide uninterrupted driving forces for Zn epitaxial growth.

Changes:

Supplementary Fig. 17. Schematic diagram of lattice match between substrate crystal and overgrowth crystal. Epitaxial substrate with (a) single crystalline orientation and (b) non-single crystalline orientation.

Furthermore, this theory is applicable to a substrate crystal with a single crystalline orientation (**Supplementary Fig. 17a**). The presence of even a small amount of other Zn textures in the Zn(0002) metal electrode induces dislocation at the epitaxial interface (**Supplementary Fig. 17b**), causing lattice distortion. Beyond the critical thickness for epitaxial growth, subsequently deposited crystals display a disordered crystalline orientation, indicating the failure of the homoepitaxial mechanism.

(In the Supporting Information, Page S13)

3. The authors report that even with ZnSO₄, approximately 30% of orientations other than (0002) exist based on calculations. This discrepancy, considering the significant theoretical value, necessitates an explanation for how 100% (0002) texture is attainable in actual experimental conditions.

Response:

Thank you very much for the reviewer's question. In theoretical calculations, the results often guide research trend and probability. Wulff structure are computationally simulated by considering variations in surface energies among distinct crystal planes, emphasizing the probability of exposure for each crystal plane. Furthermore, the construction of a three-dimensional Wulff structure becomes unfeasible if only a single crystal plane is exhibited. Building this structure involves assembling multiple facets to create a three-dimensional form. This is the reason why the (0002) crystal plane constitutes only 70% in the ZnSO₄-based solution.

4. Why do the Zn(CH₃COO)₂ and ZnCl₂-based solutions, excluding the case that used ZnSO₄, exhibit overall XRD peak shifts in supplementary Fig.4?

Response:

We greatly appreciate the reviewer's insightful question. In comparison to the Zn electrode prepared from a ZnSO₄ solution, the XRD peaks of the Zn electrodes electrodeposited from Zn(CH₃COO)₂ and ZnCl₂-based solutions exhibit a rightward shift of 0.36°. According to the Bragg equation ($2d \sin \theta = \lambda$), an increase in θ leads to a reduced interplanar spacing (d), indicating lattice contraction. Typically observed in metal electrodes, this phenomenon is attributed to two factors: a) the presence of alloying elements with smaller atomic radii in the electrodeposition layer, an inapplicable factor in this experiment; b) the existence of compressive stress in the electrodeposited sample causing lattice contraction, resulting in a rightward shift of XRD peaks—this experiment falls into this category. The Zn metal electrodes deposited from Zn(CH₃COO)₂ and ZnCl₂-based solutions exhibit a disordered crystalline orientation, inducing internal compressive stress and causing the observed rightward peak shift. Consequently, we have made adjustments and modifications to the supplementary file in response to these findings.

Changes: Furthermore, when compared to the Zn electrode prepared from a ZnSO₄ electrolyte, the XRD peaks of the Zn electrodes electrodeposited from Zn(CH₃COO)₂ and ZnCl₂-based electrolytes exhibited a rightward shift of 0.36°. According to the Bragg equation ($2d \sin \theta = \lambda$), an increase in θ

results in a reduced interplanar spacing (d), indicating lattice contraction. Consequently, the observed rightward peak shift of Zn metal electrodes deposited from $\text{Zn}(\text{CH}_3\text{COO})_2$ and ZnCl_2 -based electrolytes further demonstrate their disordered crystalline orientation. (In the Supporting Information, Page S6)

5. In Supplementary Figure 19, even considering that side reactions may be relatively severe at low current densities, the performance of the control group seems to be significantly lower compared to the reported AZIBs. Please provide a detailed explanation on this discrepancy.

Response: We express gratitude for the reviewer's insightful question; this is our oversight. In our data selection process, for the purpose of enhancing comparative distinctiveness, we initially chose data from a commercial Zn (com-Zn) electrode exhibiting relatively subpar performance. This particular battery may have experienced electrolyte leakage during its cycling process, which could have resulted in a substantial increase in polarization voltage. Consequently, we have revised the data from other com-Zn symmetric cells (Supplementary Fig. 20), replacing the prior dataset to ensure more reasonable data representation.

Changes: $\text{Zn}(0002)||\text{Zn}(0002)$ cells show superior rate performance in comparison with bare $\text{Zn}||\text{bare Zn}$ cells. When enhancing the current densities from 0.2 to 5 mA cm^{-2} and then decreasing to 0.2 mA cm^{-2} , $\text{Zn}(0002)||\text{Zn}(0002)$ cells still present stable voltage profiles with low overpotential of 37, 41, 46, 62, 99 and 36 mV, at 0.2, 0.5, 1, 2, 5, 0.2 mA cm^{-2} , respectively. (In the Supporting Information, Page S16)

Supplementary Fig. 20 | Rate performances for the $\text{com-Zn}||\text{com-Zn}$ and $\text{Zn}(0002)||\text{Zn}(0002)$ cells at various current densities and a capacity of 2 mAh cm^{-2} .

6. The manuscript predominantly focuses on high-rate performances. While high-rate performance is indeed an advantage of AZIBs, the issue of severe side-reactions at low rates is significant in this system. Given the authors' claim regarding the substantial reduction of corrosion and side reactions through the exposure of (0002) facets of Zn, it is essential to include electrochemical performance data at low current densities for both the Zn symmetric cell and full cell configuration. This information would provide a more comprehensive understanding of the electrode's behavior across a range of operating conditions.

Response: We are grateful for the reviewer's questions. It was an oversight on our part to only consider the propensity of Zn metal electrodes to form dendrites at high rates, neglecting the exacerbating of corrosion reactions at lower rates. To further illustrate the superior corrosion resistance of the Zn(0002) metal electrode, we have presented its comprehensive performance in both full cell and symmetric cell at low current densities. It can be observed that under a lower current density of 2 mA cm^{-2} , the Zn(0002) symmetric cell can stably cycle for 1800 h, and the assembled Zn(0002)||NVO full cell can stably cycle 600 cycles at a current density of 0.5 A g^{-1} , with a capacity retention rate as high as 98%. In comparison to the com-Zn metal electrode, the Zn(0002) metal electrode exhibits exceptional corrosion resistance performance.

Changes:

Fig. 5 | Electrochemical performances and practical application of aqueous Zn||NVO cells with the Zn(0002) metal anode. a Cyclic capability of the Zn||NVO coin cells using com-Zn and Zn(0002) anodes at different current densities (0.5 A g^{-1} and 5 A g^{-1}).

Supplementary Fig. 19 | Long-term galvanostatic cycling of Zn||Zn symmetric cells using com-Zn and Zn(0002) electrodes with a current density of (a) 2 mA cm^{-2} , (b) 5 mA cm^{-2} and (c) 10 mA cm^{-2} , respectively.

The com-Zn||NVO cells exhibited rapid capacity decay, while the Zn(0002)||NVO cells delivered cycling stability of up to 600 and 3500 cycles with a capacity retention of up to 98% at 0.5 A g^{-1} and 5 A g^{-1} , respectively (**Fig. 5a**). (In the Manuscript, Page 12)

Under a fixed capacity of 2 mAh cm^{-2} , the Zn(0002) symmetric cells can stably cycle for 1800 h (2 mA cm^{-2}), 3200 h (5 mA cm^{-2}) and 550 h (10 mA cm^{-2}), respectively (**Supplementary Fig. 19a-c**). In comparison to the com-Zn metal electrode, the Zn(0002) metal electrode exhibits exceptional corrosion resistance performance. (In the Supporting Information, Page S15)

7. Numerous instances of missing explanations, typos, and errors are present throughout the manuscript. For example, critical concepts, such as δ in the equation of critical thickness ($\delta = \frac{b}{f} \left(\frac{f_s}{f} \right)^2$), require proper clarification. Additionally, in Supplementary Fig.13, there are incorrect descriptions for (b) and an omission for (c). Furthermore, in Supplementary Fig.29, contrary to the authors' assertion, the SEM image of (a) com-Zn appears to be more intact and uniform than (b) Zn(0002) metal anode. These instances, along with others, need careful correction and clarification for the manuscript to meet the required standard.

Response: We deeply appreciate the thorough review conducted by the peer reviewer. We have comprehensively addressed the errors identified in both the main text and supplementary materials. Once again, thank you for your patient review.

Changes:

The t_c for the overgrowth crystal can be related to the lattice mismatch by $t_c = b(f_s/f)^2$, where f_s represents the stability threshold in the one-dimensional model of the epitaxial interface, f is only dependent on the lattice parameter (a and b) of the substrate and the epitaxial layer ($f = (b - a)/a$)^{25,26}. (In the Manuscript, Page 3)

Supplementary Fig. 13 | SEM images of (a) com-Zn, (b) IMS-Zn(0002) and (c) Zn(0002) electrodes after stripping from 5 to 30 min (scale bar, 20 μm). Current density, $J = 4 \text{ mA cm}^{-2}$. (In the

Supplementary Fig. 30 | Top-view SEM images of (a) com-Zn and (b) Zn(0002) metal anodes in Zn||NVO cells after 1000 cycles. (In the Supporting Information, Page S23)

Finally, we would like to express our gratitude to the reviewers. Engaging with their queries has deepened our understanding of the central ideas presented in this article. Additionally, the reviewers brought to light certain issues that had previously escaped our attention, for which we are truly grateful.

REVIEWERS' COMMENTS

Reviewer #2 (Remarks to the Author):

Having carefully assessed the final version of the manuscript, I appreciate the authors' dedication in addressing the queries raised during the revision process. The revisions made have significantly improved the clarity and coherence of the paper, and the authors have provided comprehensive responses to all questions posed. Based on the thorough revisions undertaken, I believe this manuscript is now suitable for publication in this journal.